# Computational and robotic modeling reveal parsimonious combinations of interactions between individuals in schooling fish

Liu Lei[1,2], Ramón Escobedo[2], Clément Sire[3], Guy Theraulaz[2]*

**1** University of Shanghai for Science and Technology, Shanghai, China, **2** Centre de Recherches sur la Cognition Animale, Centre de Biologie Intégrative, Centre National de la Recherche Scientifique (CNRS), Université de Toulouse—Paul Sabatier (UPS), Toulouse, France, **3** Laboratoire de Physique Théorique, CNRS and Université de Toulouse – Paul Sabatier, Toulouse, France

* guy.theraulaz@univ-tlse3.fr

**Data Availability Statement:** The three sets of data corresponding to i) the fish experiments (H. rhodostomus), ii) the numerical simulations of the model, and iii) the experiments with the robotic

## Abstract

Coordinated motion and collective decision-making in fish schools result from complex interactions by which individuals integrate information about the behavior of their neighbors. However, little is known about how individuals integrate this information to take decisions and control their motion. Here, we combine experiments with computational and robotic approaches to investigate the impact of different strategies for a fish to interact with its neighbors on collective swimming in groups of rummy-nose tetra (*Hemigrammus rhodostomus*). By means of a data-based agent model describing the interactions between pairs of *H. rhodostomus* (Calovi *et al.*, 2018), we show that the simple addition of the pairwise interactions with two neighbors quantitatively reproduces the collective behavior observed in groups of five fish. Increasing the number of interacting neighbors does not significantly improve the simulation results. Remarkably, and even without confinement, we find that groups remain cohesive and polarized when each agent interacts with only one of its neighbors: the one that has the strongest contribution to the heading variation of the focal agent, dubbed as the "most influential neighbor". However, group cohesion is lost when each agent only interacts with its nearest neighbor. We then investigate by means of a robotic platform the collective motion in groups of five robots. Our platform combines the implementation of the fish behavioral model and a control system to deal with real-world physical constraints. A better agreement with experimental results for fish is obtained for groups of robots only interacting with their most influential neighbor, than for robots interacting with one or even two nearest neighbors. Finally, we discuss the biological and cognitive relevance of the notion of "most influential neighbors". Overall, our results suggest that fish have to acquire only a minimal amount of information about their environment to coordinate their movements when swimming in groups.

platform are available on Figshare at:10.6084/m9.
figshare.11858379.

**Funding:** L.L. was supported by a grant from the
Natural Science Foundation of Shanghai under
grant No. 17ZR1419000. This study was supported
by grants from the Centre National de la Recherche
Scientifique (CNRS) and University Paul Sabatier
(project Dynabanc). The funders had no role in
study design, data collection and analysis, decision
to publish, or preparation of the manuscript.

**Competing interests:** The authors have declared
that no competing interests exist.

## Author summary

How do fish integrate and combine information from multiple neighbors when swimming in a school? What is the minimum amount of information about their environment needed to coordinate their motion? To answer these questions, we combine experiments with computational and robotic modeling to test several hypotheses about how individual fish could integrate and combine the information on the behavior of their neighbors when swimming in groups. Our research shows that, for both simulated agents and robots, using the information of two neighbors is sufficient to qualitatively reproduce the collective motion patterns observed in groups of fish. Remarkably, our results also show that it is possible to obtain group cohesion and coherent collective motion over long periods of time even when individuals only interact with their most influential neighbor, that is, the one that exerts the most important effect on their heading variation.

## Introduction

One of the most remarkable characteristics of group-living animals is their ability to display a wide range of complex collective behaviors and to collectively solve problems through the coordination of actions performed by the group members [1–3]. It is now well established that these collective behaviors are self-organized and mainly result from local interactions between individuals [4, 5]. Thus, to understand the mechanisms that govern collective animal behaviors, we need to decipher the interactions between individuals, to identify the information exchanged during these interactions and, finally, to characterize and quantify the effects of these interactions on the behavior of individuals [6, 7]. There exists today a growing body of work that brought detailed information about the direct and indirect interactions involved in the collective behaviors of many animal groups, especially in social insects such as ants [8–11] and bees [12, 13].

Recently, we introduced a new method to disentangle and reconstruct the pairwise interactions involved in the coordinated motion of animal groups such as fish schools, flocks of birds, and human crowds [14, 15]. This method leads to explicit and concise models which are straightforward to implement numerically. It still remains an open and challenging problem to understand how individuals traveling in groups combine the information coming from their neighbors to coordinate their own motion.

To answer this question, one first needs to identify which of its neighbors an individual interacts with in a group, *i.e.*, which are its influential neighbors. For instance, does an individual always interact with its nearest neighbors, and how many? Most models of collective motion in animal groups generally consider that each individual is influenced by all the neighbors located within some spatial domain centered around this individual [16, 17]. This is the case in particular of the Aoki-Couzin model [18, 19] and the Vicsek model [20]. In the latter, each individual aligns its direction of motion with the average direction of all individuals that are located within a fixed distance in its neighborhood. Other models, more directly connected to biological data, consider that the interactions between individuals are topological and that the movement of each individual in the group only relies on a finite number of neighbors. This is in particular the case for the work on starling flocks [21, 22] and on barred flagtails (*Kuhlia mugil*) [23]. In golden shiners (*Notemigonus crysoleucas*), another work has sought to reconstruct the visual information available to each individual fish during collective evasion maneuvers [24]. In this species, it has been shown that the transmission of behavior in a school was best described by a model in which the response probability of a fish depends on the fraction

of active neighbors perceived by that fish. However, because of the cognitive load that is required for an individual to constantly monitor the movements of a large number of neighbors, it has been suggested that animals may focus their attention on a small subset of their neighbors [25–27]. In a previous work, we found experimental evidence that supports this assumption. In groups of rummy nose tetras (*Hemigrammus rhodostomus*) performing collective U-turns, we found that, at any time, each fish pays attention to only a small subset of its neighbors, typically one or two, whose identity regularly changes [28]. However, we still do not know if the same pattern of interaction holds true when fish are schooling, *i.e.*, when individuals are moving together in a highly polarized manner and not performing some collective maneuver.

Once the influential neighbors of a focal fish have been identified, one must then understand how this individual combines the information about the behavior of these neighbors. The most common assumption is that animals respond by averaging pairwise responses to their neighbors (with added noise) [16–18]. However, existing work shows that the integration of information might be much more complex. In golden shiners, Katz *et al.*[29] have shown that the combined effect of two neighbors on a fish response is close to averaging for turning, but somewhere between averaging and adding for speed adjustments. This observation brings us back to an often neglected factor which is the impact of the physical constraints imposed on a fish movement by their body. Fish mainly achieve collision avoidance through the control of their speed and orientation at the individual level. However, existing models seldom treat collision avoidance in a physical way and most models assume that individuals move at a constant speed [6]. This is the main reason why these models cannot be directly implemented in real physical robotic systems [30].

To better understand how individuals integrate and combine interactions with their neighbors in a group of moving animals, we first analyze the dynamics of collective movements in groups of five *H. rhodostomus* moving freely in a circular tank. Then, we investigate different strategies for combining pairwise interactions between fish and analyze their impact on collective motion. To do that, we use the data-driven computational model developed by Calovi *et al.*[14] that describes the interactions involved in the coordination of burst-and-coast swimming in pairs of *H. rhodostomus*, and a robotic platform that also allows us to investigate the impact of direction and speed regulation, and of collision avoidance. Finally, we compare the predictions of the computational and robotics models with the experiments conducted under the same conditions with groups of fish.

## Results

We collect three sets of data corresponding to *i*) our experiments with $N = 5$ fish (*H. rhodostomus*), *ii*) our numerical simulations of the model derived in [14], and *iii*) our experiments using the robotic platform with $N = 5$ robots (see Fig 1, S1 and S2 Videos), from which we extract the trajectories of each individual (S3 Video). We characterize the collective behavior of fish, agents, and robots by means of five main quantities:

- the group cohesion $C(t)$ at time $t$, which characterizes the effective radius of the group, and hence its compactness;

- the group polarization $P(t)$, which quantifies the coordination of the headings of the individuals ($P(t) = 1$, if all individuals are perfectly aligned; $P \sim 1/\sqrt{N}$, if the $N$ individuals have uncorrelated headings, $P$ becoming small only for large group size $N$, but being markedly lower than 1 for any $N \geq 5$);

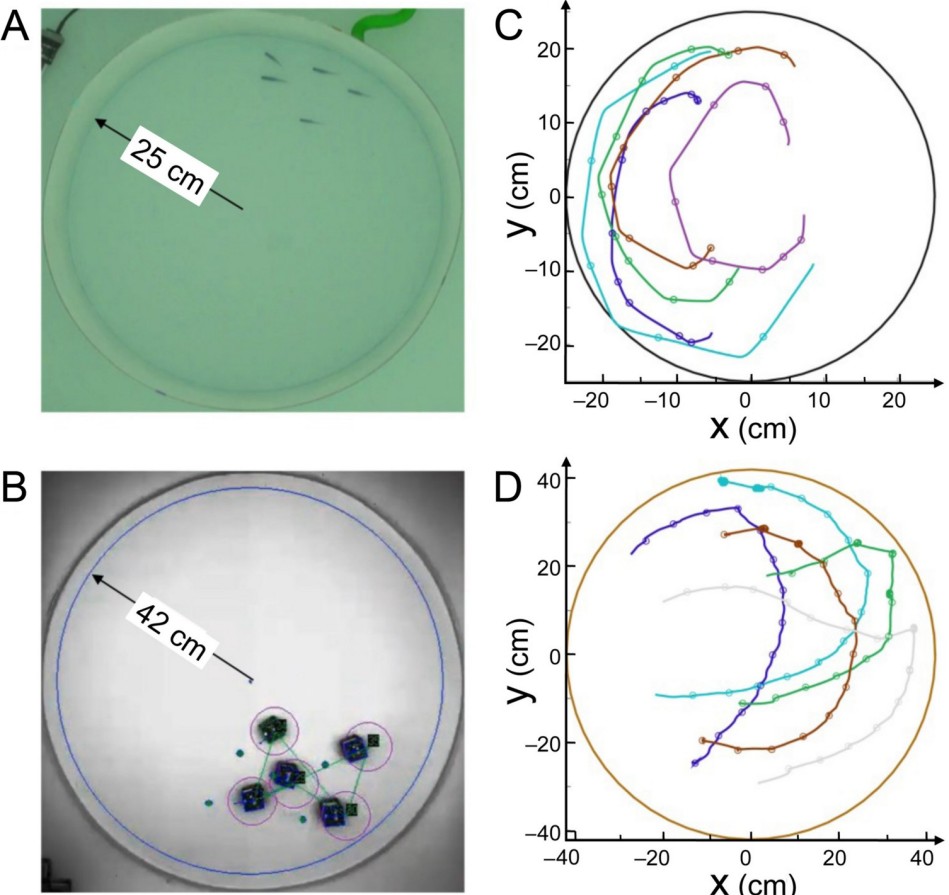

**Fig 1. Experimental setups and tracking.** (A) Experiments with 5 fish swimming in a tank of radius $R_{\text{fish}}$ = 25 cm. (B) 5 robots running in a platform of radius $R_{\text{robot}}$ = 42 cm. (C) Individual fish trajectories over 4 seconds. The circles represent the onset of bursts, when speed is minimum. (D) Individual trajectories in one robotic experiment over 24 seconds. The circles indicate the decisions of the robots to select a new target place, when individual speed is minimum.

- the distance $r_w^B(t)$ of the barycenter $B$ of the group from the wall of the tank, which is only small compared to the radius of the tank if individuals move together *and* along the wall of the tank;

- the relative orientation $\theta_w^B(t)$ of the barycenter of the group with respect to the wall of the tank, which in particular characterizes whether the group is collectively swimming parallel to the wall of the tank (then, $|\theta_w^B| \approx 90°$);

- the counter-milling index $Q(t)$, which measures the relative direction of rotation of individuals inside the group (around the barycenter) with respect to the direction of rotation of the group around the center of the tank (see S4 Video).

The Materials and Methods section and Figs 2 and 3 provide the precise mathematical definition of these quantities. Moreover, we used the Hellinger distance (see Materials and methods) between two probability distribution functions (PDF) in order to quantify the

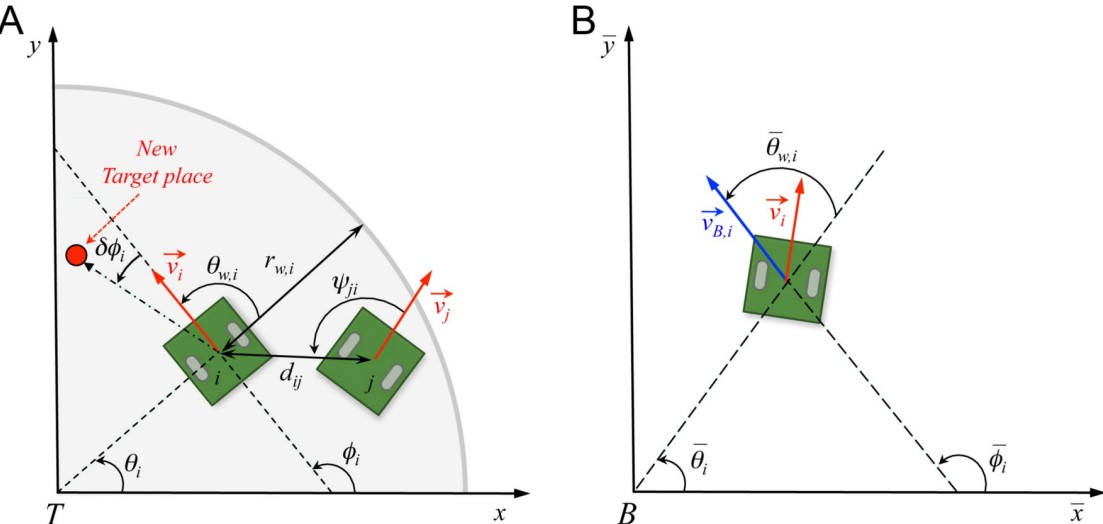

**Fig 2. Angles and reference systems.** (A) Distances, angles, and velocity vectors of agents $i$ and $j$ in the absolute reference system centered in $T(0, 0)$. Positive values of angles are fixed in the anticlockwise direction. $\theta_i$ is the position angle of agent $i$ with respect to $T$ and the horizontal line; $r_{w,i}$ is the distance of agent $i$ from the nearest wall; $\phi_i$ is the heading angle of agent $i$, determined by its velocity vector $\vec{v}_i$; $\theta_{w,i}$ is the relative angle of agent $i$ with the wall; $d_{ij}$ is the distance between agents $i$ and $j$; $\psi_{ji}$ is the viewing angle with which agent $i$ perceives agent $j$, *i.e.*, the angle between the velocity of $i$ and the vector $\vec{ij}$ (we show the angle $\psi_{ji} \neq \psi_{ij}$ with which $j$ perceives $i$, for the sake of readability of the figure); $\phi_{ij} = \phi_j - \phi_i$ is the difference of heading between agents $i$ and $j$, and $\delta\phi_i$ is the variation of heading of agent $i$. (B) Relative reference system centered in the barycenter of the group $B(x_B, y_B)$. Relative variables are denoted with a bar. Angle $\bar{\theta}_{w,i} = \bar{\phi}_i - \bar{\theta}_i$ is the angle of incidence of the relative speed of agent $i$ with respect to a circle centered in $B$.

(dis)similarity between PDF obtained in fish experiments and the corresponding PDF obtained in the fish model (see Table 1) and in robot experiments (see Table 2).

*H. rhodostomus* presents a burst-and-coast swimming mode, where a fish suddenly accelerates along a new direction ("kick"; see Fig 1B, and S1 and S3 Videos) and then glides passively until the next kick, along an almost straight line, a gliding phase during which the speed approximately decays exponentially [14]. The fish model derived in [14] explicitly implements this swimming mode and returns as the main information the new heading direction of the focal fish after each kick, which is controlled by its environment (wall of the tank, another fish). The interaction between a fish and the wall, and the interaction between two fish have been precisely extracted from actual experiments with *H. rhodostomus* [14]. The original procedure for extracting the interactions introduced in [14] exploited a large data set of $\sim$300000 kicks for one-fish trajectories (in tanks of 3 different radii) and $\sim$200000 kicks for two-fish trajectories, amounting effectively to a total of 70 hours of exploitable data. The measured interactions were then directly implemented in the model, which is hence not just a phenomenological model with mere guessed, albeit reasonable, interactions. Note however that the analysis in [14] does not provide any insight about how these interactions are combined in groups of more than two fish.

The interaction between two fish was shown to be a combination of a repulsive (at short distance of order 1 BL—body length) and a long-range (in particular, compared to zebrafish [15]) attractive interaction at larger distance, and of an alignment interaction which tends to make the two fish align their heading direction. The attraction and alignment interaction functions determine the new heading angle of the focal fish in terms of the instantaneous relative state of the two fish, characterized by the distance between them, the viewing angle with which the neighbor is perceived by the focal fish, and their relative orientation (see Fig 2). The

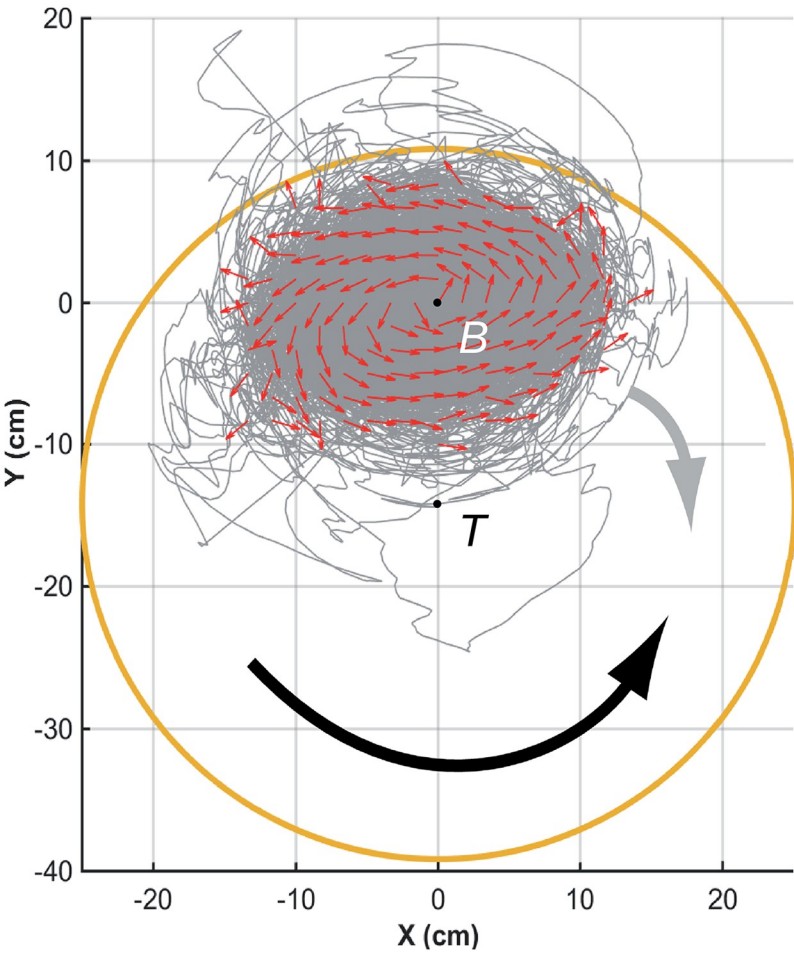

**Fig 3. Counter-milling in fish experiments.** Individual fish (small red arrows) turn counter-clockwise (CCW) around their barycenter, here located at $B(0, 0)$, while the fish group rotates clockwise (CW) around the center of the tank, located at $T(0, -14)$ in the reference system of the barycenter. Red arrows (of same length) denote relative fish heading, gray lines denote relative trajectories, and large orange circle denotes the average relative position of the border of the tank. The wide black arrow shows the direction of rotation of individual fish with respect to $B$ (CCW), opposite to the wide gray arrow showing the direction of rotation of the group with respect to $T$ (CW).

additional change in heading angle due to the repulsive interaction between a fish and the wall of a circular tank is expressed in terms of the distance and relative angle of the fish to the wall (see Fig 2). Finally, in addition to the fish-wall and fish-fish interactions, the change in heading angle includes a stochastic contribution describing the spontaneous fluctuations in the motion of the fish. In [14], the model was shown to quantitatively reproduce many fine measurable quantities in one-fish and two-fish experiments, ultimately producing a very precise description of the motion of one or two fish. For the sake of completeness, the model and its fish-wall and fish-fish interaction functions are summarized in the Materials and Methods section (Eqs (4)–(15); see [14] for a more detailed description and justification of the model; see [14, 15] for the extraction procedure of the interactions).

When more than two fish are swimming in the tank ($N > 2$), the social pairwise interactions must be combined. In the framework of the fish model, it is natural to assume that the heading angle change of a focal fish is the sum of the pairwise contributions of *some* of its

**Table 1. Model simulations vs fish experiments.** Distance $D$(Fish|Model) between the probability distribution function (PDF) of the 5 observables used to quantify the collective motion in the fish model and the corresponding PDF obtained in fish experiments. We list the results for the 3 different interaction strategies implemented in the fish model and the associated value of $k$ for the number of interacting neighbors. The last column $\langle$All$\rangle$ corresponds to the average of the 5 corresponding distances, an arbitrary but reasonable global quantifier to assess the overall agreement of a given condition with the results of fish experiments. For $k = 1$, the MOST INFLUENTIAL strategy gives significantly better results than the two other strategies and already leads to a fair agreement with fish experiments.

| STRATEGY | | $C$ | $P$ | $r_w^B$ | $\theta_w^B$ | $Q$ | $\langle$**All**$\rangle$ |
|---|---|---|---|---|---|---|---|
| | $k = 0$ | 0.909 | 0.532 | 0.341 | 0.145 | 0.023 | 0.390 |
| NEAREST | $k = 1$ | 0.369 | 0.178 | 0.034 | 0.041 | 0.003 | 0.125 |
| | $k = 2$ | 0.065 | 0.049 | 0.032 | 0.033 | 0.020 | 0.040 |
| | $k = 3$ | 0.013 | 0.026 | 0.027 | 0.032 | 0.037 | 0.027 |
| RANDOM | $k = 1$ | 0.310 | 0.223 | 0.095 | 0.068 | 0.009 | 0.141 |
| | $k = 2$ | 0.061 | 0.103 | 0.037 | 0.059 | 0.037 | 0.059 |
| | $k = 3$ | 0.012 | 0.062 | 0.028 | 0.048 | 0.038 | 0.038 |
| MOST INFLUENTIAL | $k = 1$ | 0.078 | 0.150 | 0.067 | 0.048 | 0.006 | 0.070 |
| | $k = 2$ | 0.011 | 0.051 | 0.025 | 0.080 | 0.033 | 0.040 |
| | $k = 3$ | 0.016 | 0.038 | 0.027 | 0.042 | 0.036 | 0.032 |
| | $k = 4$ | 0.014 | 0.042 | 0.024 | 0.044 | 0.030 | 0.031 |

$N - 1$ neighbors. The resulting interaction thus depends on two factors: the *number $k$ of considered neighbors and the *strategy* to select them.

We explore three different strategies of interaction between individuals and their neighbors in groups of size $N = 5$, comparing actual fish experiments with the resulting fish model and the robotic platform. In the latter, the robots are programmed with the fish model and a control procedure to resolve collisions. The first strategy is based on the *distance*, so that individuals interact with their $k$ nearest neighbors, with $k = 1, 2, 3$. The second strategy is a *random* strategy, where the $k$ neighbors are randomly sampled among the other $N - 1$ individuals. Finally, the third strategy is based on the *influence*, defined below, where the $k$ selected neighbors are those having the largest influence on the focal individual (as determined by the precise two-fish model of [14]). We also study the cases where there is no interaction between individuals ($k = 0$), and where each individual interacts with all its neighbors ($k = 4$).

**Table 2. Collective robotics experiments vs fish experiments.** Distance $D$(Fish|Robots) between the probability distribution function (PDF) of the 5 observables used to quantify the collective motion of the robots and the corresponding PDF obtained in fish experiments. We list the results for the 3 different interaction strategies implemented in the fish model and the associated value of $k$ for the number of interacting neighbors. The last column $\langle$All$\rangle$ corresponds to the average of the 5 corresponding distances, an arbitrary but reasonable global quantifier to assess the overall agreement of a given condition with the results of fish experiments. For $k = 1$, the MOST INFLUENTIAL strategy gives significantly better results than the two other strategies and already leads to a fair agreement with fish experiments.

| STRATEGY | | $C$ | $P$ | $r_w^B$ | $\theta_w^B$ | $Q$ | $\langle$**All**$\rangle$ |
|---|---|---|---|---|---|---|---|
| | $k = 0$ | 0.604 | 0.561 | 0.238 | 0.114 | 0.170 | 0.337 |
| NEAREST | $k = 1$ | 0.418 | 0.486 | 0.158 | 0.070 | 0.239 | 0.274 |
| | $k = 2$ | 0.111 | 0.249 | 0.063 | 0.042 | 0.093 | 0.112 |
| | $k = 3$ | 0.066 | 0.039 | 0.083 | 0.036 | 0.026 | 0.05 |
| RANDOM | $k = 1$ | 0.140 | 0.343 | 0.040 | 0.107 | 0.065 | 0.139 |
| | $k = 2$ | 0.019 | 0.141 | 0.035 | 0.080 | 0.029 | 0.061 |
| | $k = 3$ | 0.056 | 0.063 | 0.095 | 0.042 | 0.025 | 0.056 |
| MOST INFLUENTIAL | $k = 1$ | 0.045 | 0.089 | 0.050 | 0.042 | 0.011 | 0.047 |
| | $k = 2$ | 0.028 | 0.050 | 0.031 | 0.088 | 0.024 | 0.044 |
| | $k = 4$ | 0.078 | 0.080 | 0.040 | 0.053 | 0.038 | 0.058 |

The *influence $\mathcal{I}_{ij}(t)$ of a neighbor $j$ on a focal individual $i$ at time $t$* is defined as the intensity of the contribution of this neighbor $j$ to the instantaneous heading variation of the focal individual $i$, as given by the firmly tested two-fish model of [14]. The influence $\mathcal{I}_{ij}(t)$ depends on the relative state of the neighbor $j$ with respect to the focal individual $i$, determined by the triplet $(d_{ij}, \psi_{ij}, \phi_{ij})$, where $d_{ij}$ is the distance between individuals $i$ and $j$, $\psi_{ij}$ is the viewing angle with which $i$ perceives $j$ (*i.e.*, the angle between the velocity of $i$ and the vector $\vec{ij}$), and $\phi_{ij}$ is the difference of their heading angles, a measure of the alignment between $i$ and $j$ (see Fig 2). The influence $\mathcal{I}_{ij}(t)$ is evaluated at each kicking time of individual $i$ by means of the analytical expressions of the pairwise interaction functions derived in [14] for fish swimming in pairs, according to Eq (9) in the Materials and Methods section.

To prevent cognitive overload, a reasonable assumption is that individual fish filter the information from their environment and thus limit their attention to a small set of their most salient neighbors [25–27] (to be followed; or to be avoided, by moving away or by aligning their headings), making the notion of most influential neighbors quite natural.

The model for $N > 2$ agents thus proceeds as follows: at the time when the agent performs a new kick, its change in heading angle is calculated by adding the effects of the wall and the spontaneous noise to the effects of the $k$ neighbors selected among the other $N - 1$ individuals according to one of the three strategies presented above:

- by calculating the instantaneous distance between the focal individual $i$ and each of its $N - 1$ neighbors and selecting the $k$ nearest neighbors (strategy 1; NEAREST);

- by randomly sampling $k$ individuals among the $N - 1$ neighbors of $i$ (strategy 2; RANDOM);

- by calculating the instantaneous influence $\mathcal{I}_{ij}(t)$ for each neighbor $j$ of $i$ and selecting the $k$ neighbors with the largest influence (strategy 3; MOST INFLUENTIAL).

The strategy is thus characterized by the number $k$ of neighbors taken into account in the social interaction and the criterion used to select them (NEAREST, RANDOM, or MOST INFLUENTIAL). The strategy remains unchanged along the whole simulation. However, the identity of the neighbors selected to interact with a given agent can change from one kick to another, and must be updated at each kicking time of this agent. For instance, when using the NEAREST strategy with $k = 2$ in a group of $N = 5$ agents, the agents taken into account in the social interaction in the $n$-th kick of agent 1 can be the agents 2 and 3, and the agents 4 and 3 in its $(n + 1)$-th kick. In order to select these $k$ neighbors at a specific kick, the $N - 1$ agents must be sorted according to the criterion corresponding to the strategy used in the simulation. This sorting process is carried out at each kicking time of the focal agent, independently of the state (kicking or gliding) of the other agents. If $N$ is so large that the computational cost of this process becomes prohibitive, a more efficient algorithm can be implemented, such as keeping track of the agents that were selected in the most recent kicks and exploiting grid algorithms to identify neighbors.

These interaction strategies explore different ways for an individual to focus its attention on the most relevant stimuli (*i.e.*, neighbors).

## Collective behavior in fish experiments

Fish form cohesive groups with an average cohesion $C \approx 5$ cm (Fig 4). They are highly polarized, with the 5 fish swimming almost in the same direction (large peak at $P \approx 1$ in the distribution of $P$; Fig 5). In some instances, groups are observed in which one fish swims in the opposite direction to that of the other four, as shown by the small bump at $P \approx 0.6$ in Fig 5. Indeed, in this situation, the polarization is close to $P \approx |1 + 1 + 1 + 1 - 1|/5 = 0.6$. Even less

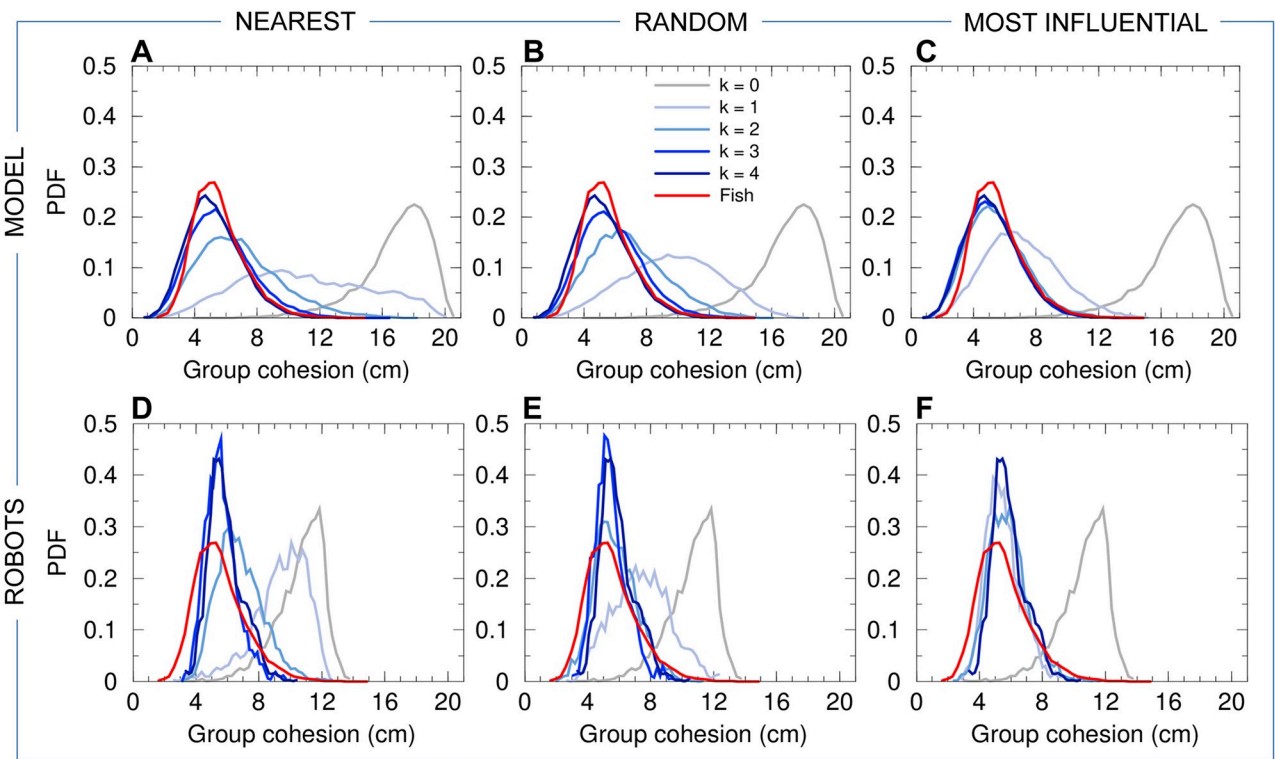

**Fig 4. Group cohesion.** Probability density functions (PDF) of the group cohesion $C$ for the experiments with 5 fish (red line in all panels), model simulations (panels ABC), and experiments with 5 robots (panels DEF), compared to the corresponding null models ($k = 0$, no interaction between individuals) in both simulations and robots (gray line in all panels). Distances have been rescaled by $\lambda_M = 0.87$ for the model simulations, and by $\lambda_R = 0.35$ for the robot experiments. The intensity of blue is proportional to the number of neighbors interacting with a focal individual (agent or robot), from $k = 1$ (light blue) to $k = 4$ (dark blue). Interaction strategies involve the $k$ NEAREST neighbors (panels AD), $k$ RANDOM neighbors (panels BE), and the $k$ MOST INFLUENTIAL neighbors (panels CF).

frequent are situations where two fish swim in the opposite direction to that of the other three, as shown by the very small bump near $P \approx |1 + 1 + 1 - 1 - 1|/5 = 0.2$. The density maps of polarization $P$ with respect to cohesion $C$ (panels labeled "FISH" in S1–S4 Figs) allow to visualize the correlations between both quantities, and will permit a comparison with the predictions of the fish model and the results of the robot experiments for the three interaction strategies considered here.

Groups of 5 fish rotate clockwise (CW) or counter-clockwise (CCW) along the tank wall for long periods and remain close to the border of the tank, the group barycenter being at a typical distance $r_w^B \approx 7$ cm from the wall (Fig 6). Therefore, the group swims almost always parallel to the nearest wall, with a relative angle to the wall of the heading of the barycenter close to $|\theta_w^B| \approx 90°$ (Fig 7). In fact, the peak in the PDF of $|\theta_w^B|$ is slightly below 90°, since the fish are more often going toward the wall than away from it [14].

We also find a collective pattern where individual fish rotate around the barycenter $B$ of the group in a direction which is *opposite* to the direction of rotation of the group around the center $T$ of the tank (see Fig 3 and S4 Video). We call this collective movement a *counter-milling behavior*, and define the instantaneous degree of counter-milling $Q(t)$ as a measure in $[-1, 1]$ of the intensity with which both rotation movements are in opposed directions (see the Materials and methods section for the precise mathematical definition of $Q(t)$ and its general interpretation). When $Q(t)<0$, the fish rotate around their barycenter $B$ in the opposite direction to

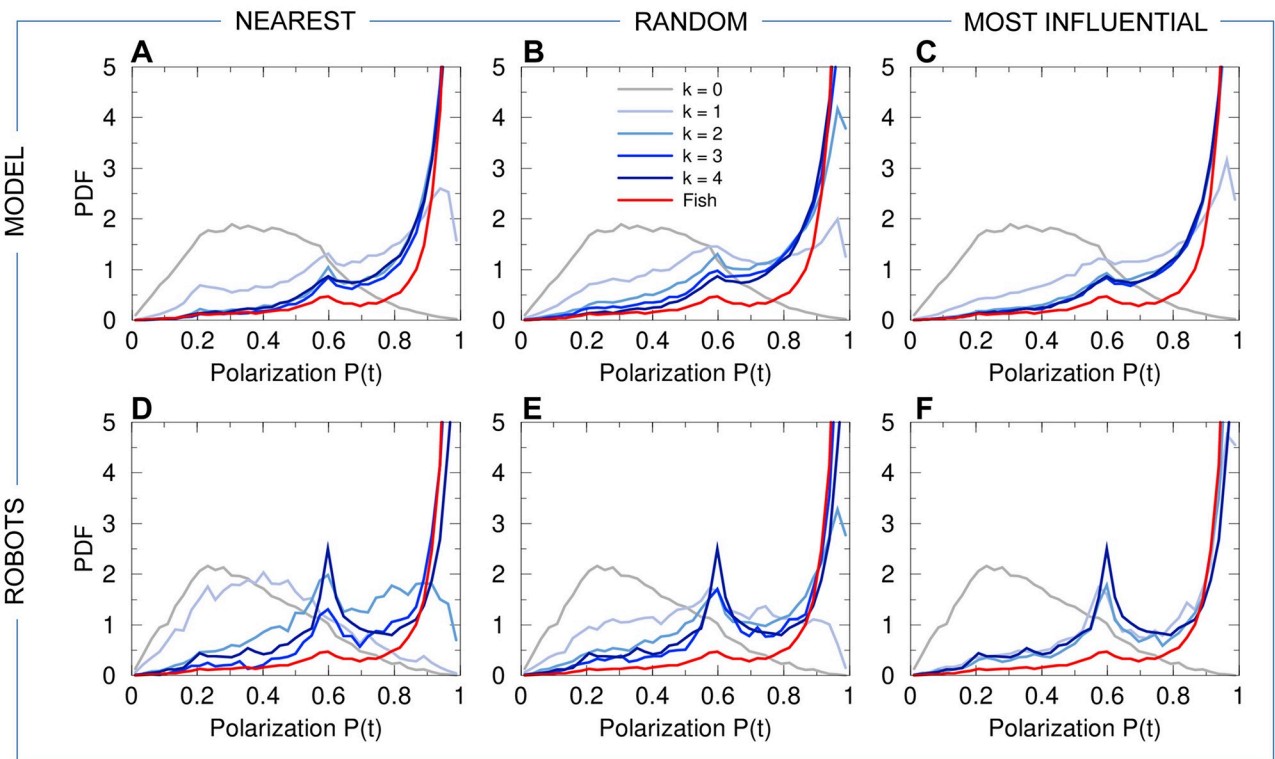

**Fig 5. Group polarization.** PDF of the group polarization *P* for fish experiments (red line in all panels), model simulations (panels ABC), and robot experiments (panels DEF), compared to the corresponding null models (*k* = 0, no interaction between individuals) in both simulations and robots (gray line in all panels). Curves for agents (fish model and robots) are in blue and gray, depending on the value of *k* (see legend in panel B). Interaction strategies involve the *k* NEAREST neighbors (panels AD), *k* RANDOM neighbors (panels BE), and the *k* MOST INFLUENTIAL neighbors (panels CF).

that of the group around *T* (*counter-milling*), while when $Q(t) > 0$, the fish rotate in the same direction around *B* as the group rotates around *T* (*super-milling*). Fig 8 shows that the fish exhibit a counter-milling behavior much more frequently than a super-milling behavior. Counter-milling behaviors result from the fact that fish located at the front of the group have to reduce their speed as they get closer to the wall of the tank. Fish located at the back of the group (that are generally farther from the wall [14]) move faster and outrun the slowing down fish, ultimately relegating them to the back of the group. This process gives rise to the rotation of individual fish around the group center, in the opposite direction to the one that the group displays around the tank (Fig 3). This collective behavior resembles a coordinated swimming by relays which is nevertheless due to simple physical constraints, as already reported on wolf-packs hunting preys moving in circles [31].

## Simulation results of the computational model

**Collective motion in a circular tank.** Panels (ABC) of Figs 4–8 show the probability distribution functions for our 5 quantifiers as obtained in numerical simulations of the fish model. The panels correspond respectively to the strategy in which agents interact with their *k* nearest neighbors (A), with *k* neighbors chosen randomly (B), and with *k* neighbors selected according to their influence on the focal agent (C). For these three strategies (NEAREST; RANDOM; MOST INFLUENTIAL), we have considered all the possible values of the number of interacting neighbors, *k* = 1, 2, 3, together with the case where there is no interaction between agents (*k* = 0) and the case where each agent interacts with every other agent (*k* = 4).

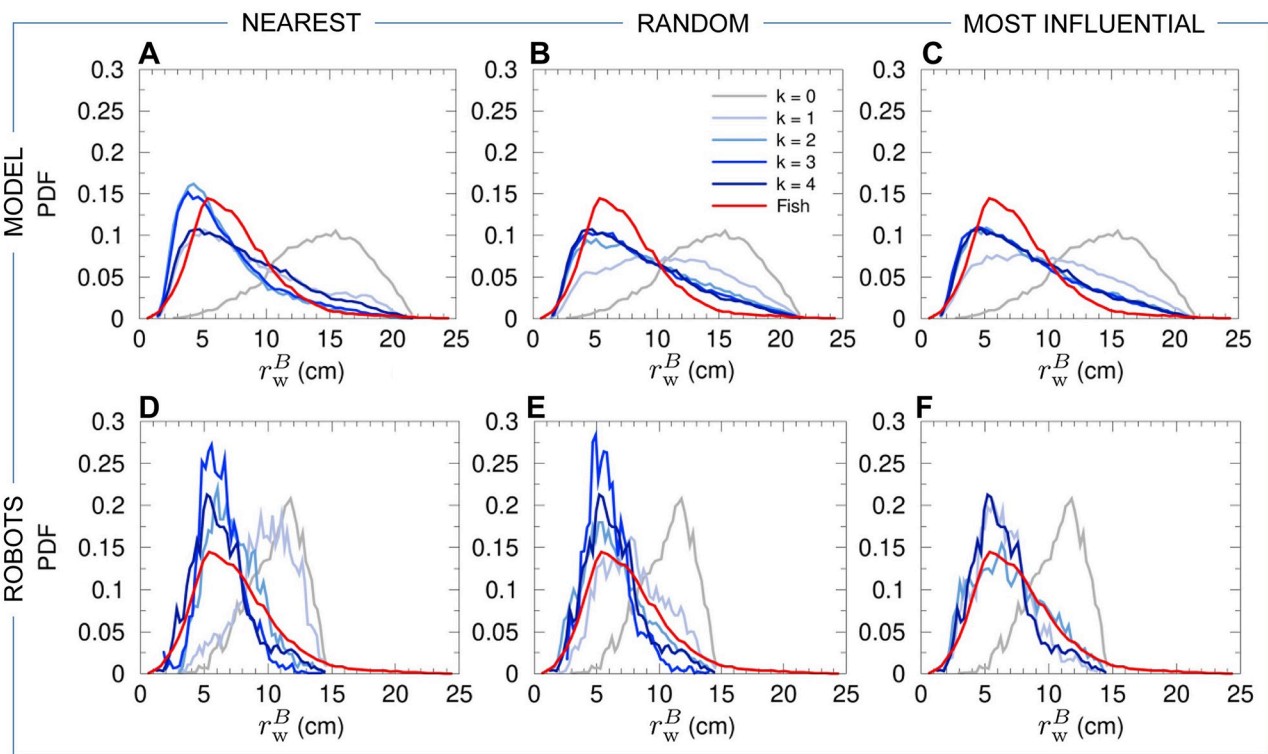

**Fig 6. Distance of the barycenter of the individuals to the wall.** PDF of the distance $r_w^B$ of the barycenter of the individuals from the wall for fish experiments (red line in all panels), model simulations (panels ABC), and robot experiments (panels DEF), compared to the corresponding null models ($k = 0$, no interaction between individuals) in both simulations and robots (gray line in all panels). Distances have been rescaled by $\lambda_M = 0.87$ for the model simulations, and by $\lambda_R = 0.35$ for the robot experiments. Curves for agents (fish model and robots) are in blue and gray, depending on the value of $k$ (see legend in panel B). Interaction strategies involve the $k$ NEAREST neighbors (panels AD), $k$ RANDOM neighbors (panels BE), and the $k$ MOST INFLUENTIAL neighbors (panels CF).

For comparison purposes, we have rescaled the distance corresponding to the model by a factor $\lambda_M = 0.87$. This value is the minimizer of the $l_1$-norm of the difference between the PDF of group cohesion for fish data, and the PDF of group cohesion for the simulation data produced by the model when using the strategy involving the $k = 2$ most influential neighbors. Noticeably, the fact that the value of $\lambda_M$ is close to 1 indicates that the model produces a quite satisfactory quantitative approximation to the data of real fish. This rescaling procedure only affects the PDF of $C$ and $r_w^B$, and not the PDF of $P$, $\theta_w^B$, and $Q$ (3 quantities invariant by a change of distance scale).

When $k = 0$, there is no interaction between agents and, as expected, one does not observe any compact group: individuals turn independently around the tank remaining close and parallel to the wall (as expected for fish swimming alone [14]). Their position and rotation direction along the walls are uncorrelated, and the individuals are scattered along the border (cohesion peaked around $C \approx 18$ cm; $r_w^B \approx 15$ cm), with an almost flat PDF for $\theta_w^B$ (random orientation of the barycenter with respect to the wall). This results in a bell-shaped probability distribution function PDF for the polarization $P$, vanishing at $P = 1$ (Figs 4–7).

For $k = 1$, whatever the strategy used to select the interacting neighbor (the nearest one; a randomly selected one; the most influential one), the dynamics immediately reveals that interactions are at play, with groups becoming more cohesive (Fig 4) and more polarized (Fig 5) than for $k = 0$. Yet, the NEAREST strategy still leads to a very broad PDF of the group cohesion $C$,

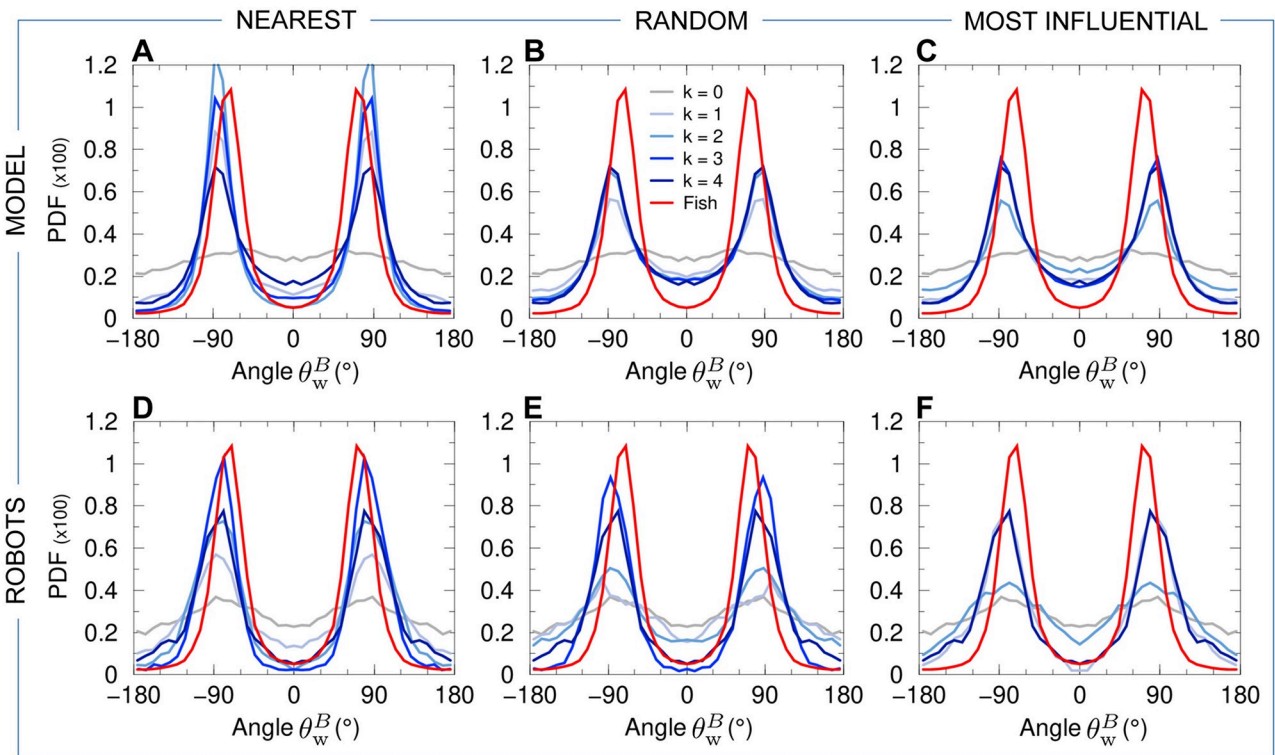

**Fig 7. Relative angle of the heading of the barycenter of the group with the wall.** PDF of the relative angle $\theta_w^B$ of the heading of the barycenter of the group with the wall for fish experiments (red line in all panels), model simulations (panels ABC), and robot experiments (panels DEF), compared to the corresponding null models ($k = 0$, no interaction between individuals) in both simulations and robots (gray line in all panels). Curves for agents (fish model and robots) are in blue and gray, depending on the value of $k$ (see legend in panel B). Interaction strategies involve the $k$ NEAREST neighbors (panels AD), $k$ RANDOM neighbors (panels BE), and the $k$ MOST INFLUENTIAL neighbors (panels CF).

with a substantial weight near the maximal value of $C \sim 20$ cm obtained for $k = 0$, indicating that the group often breaks into parts. For the RANDOM and MOST INFLUENTIAL strategies, the weight at large distance in the PDF of $C$ is absent, but the PDF are still broader than in fish experiments. As confirmed by the Hellinger distance quantifier (see Table 1 and Materials and methods), the MOST INFLUENTIAL strategy clearly leads to the sharper distribution of $C$ (peaked around $C \approx 6.5$ cm, compared to $C \approx 10$ cm for the RANDOM strategy). The next section will show that, contrary to the NEAREST strategy, the MOST INFLUENTIAL strategy with $k = 1$ can lead to compactness of the group even for larger groups ($N = 6$–70) moving in an *unbounded* domain. As for the group polarization $P$ (Fig 5), the three strategies lead to a PDF clearly peaked near $P \approx 0.9$ (and a smaller peak near $P \approx 0.6$; see above), yet certainly not as peaked near $P = 1$ as the PDF for fish experiments. Again, the MOST INFLUENTIAL strategy leads to the best agreement with fish experiments (see Table 1), although the difference between strategies is not as marked as for the group cohesion. For the three strategies, the barycenter of the group is closer to the border and moves more parallel to the wall (Figs 6 and 7). Counter-milling is obtained for the three strategies with comparable PDF (Fig 8; see also S5 Fig), quite similar to the one obtained in fish experiments (we will see that the agreement unfortunately worsens when increasing $k$; see Table 1). Polarization vs cohesion density maps confirm that the NEAREST and RANDOM strategies are insufficient to convey the necessary information to reach the degree of cohesion and polarization (and their correlation) observed in groups of fish (S1 and S2 Figs). The MOST INFLUENTIAL strategy density maps for $k = 1$ already present the main features of the fish

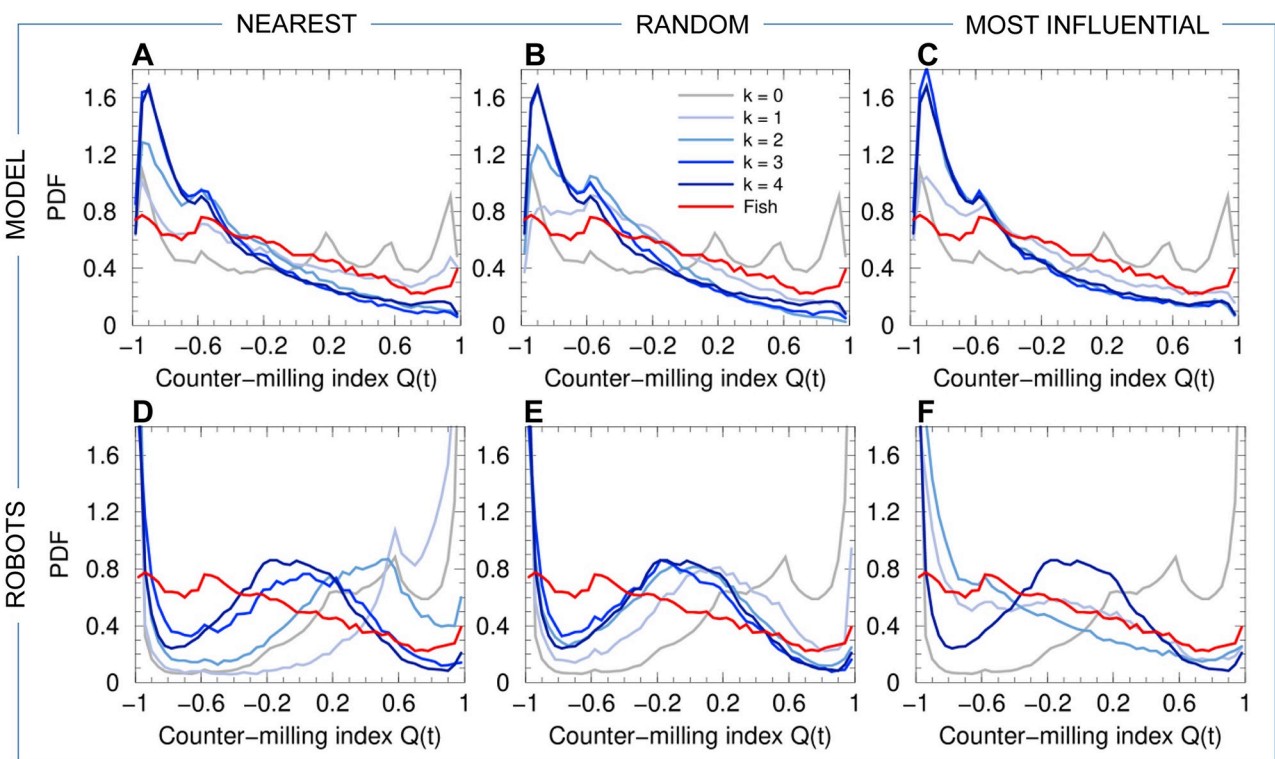

**Fig 8. Counter-milling index.** PDF of the counter-milling index $Q$ for fish experiments (red line in all panels), model simulations (panels ABC), and robot experiments (panels DEF), compared to the corresponding null models ($k = 0$, no interaction between individuals) in both simulations and robots (gray line in all panels). Curves for agents (fish model and robots) are in blue and gray, depending on the value of $k$ (see legend in panel B). Interaction strategies involve the $k$ NEAREST neighbors (panels AD), $k$ RANDOM neighbors (panels BE), and the $k$ MOST INFLUENTIAL neighbors (panels CF).

experiments, despite a still too broad spreading in the $(C, P)$ plane. Overall, for $k = 1$, the MOST INFLUENTIAL strategy gives rise to significantly better results than the NEAREST and RANDOM strategies (see Table 1).

For $k = 2$, the three strategies lead to a collective behavior in much better agreement with the fish experiments (see Table 1). In particular, the NEAREST strategy now systematically leads to compact groups, with a PDF of the group cohesion $C$ (Fig 4) similar to the one obtained for the RANDOM strategy (both peaked around $C \approx 6.5$ cm). The MOST INFLUENTIAL strategy produces a PDF in good agreement with fish experiments (both sharply peaked around $C \approx 5$ cm). The PDF of the polarization is now sharply peaked at $P = 1$ for the three strategies, with a slightly lower level of polarization for the RANDOM strategy compared to the two others (see Fig 5 and Table 1). Like in the case $k = 1$, the distance and alignment of the group with respect to the wall are better recovered for the NEAREST strategy (Figs 6 and 7; Table 1), the two other strategies leading to slightly broader PDF but much narrower compared to the case $k = 1$. The counter-milling $Q$ is enhanced for the three strategies compared to the case $k = 1$ and appears stronger than for fish experiments (Fig 8). The deterioration of the model results for the counter-milling compared to $k = 1$ and experiments suggests that the internal structure of a fish group is more rigid than predicted by the model, actual fish behaving closer to particles rotating on a vinyl record (see the interpretation of $Q$ in Materials and Methods). Compared to the case $k = 1$, where they were particularly far from the experimental maps, polarization vs cohesion density maps for the NEAREST and RANDOM strategies and $k = 2$ show a correlation between $P$ and $C$ in much better agreement with experiments (S1 and S2 Figs). The MOST

INFLUENTIAL strategy results, already fair for $k = 1$, also improve. The NEAREST strategy leads to the best agreement with experiments in the representation of S1 Fig, while the MOST INFLUENTIAL strategy leads to the best results in the representation of S2 Fig.

When interacting with $k = 3$ neighbors, the results are almost identical for the three strategies because neighbors are the same a high percentage of the time. For two (respectively, three) given strategies, the selected neighbors are exactly the same 25% of the time (respectively, 6.25%); they have at least 2 neighbors in common 75% of the time (respectively, 93.75%); there is always at least one neighbor in common. Interacting with the 3 nearest neighbors instead of 2 only improves the group cohesion (see Table 1 and Fig 4), while using the 3 most influential ones, instead of 2, does not improve significantly any of the measures, including density maps (S1 and S2 Figs). As already noted for $k = 2$, the counter-milling remains too pronounced compared to experiments for the three strategies and $k = 3$ (see Fig 8 and S5 Fig).

Finally, interacting with $k = 4$ neighbors does not significantly change the results obtained for $k = 3$ (see Figs 4–8 and Table 1).

**Collective motion of 5 agents in an unbounded domain.** The model allows us to simulate the condition where agents are swimming in an unbounded domain by removing the interaction with the wall. This condition is particularly interesting to assess the impact of the confinement of the agents due to the arena on group cohesion and polarization.

Figs 9 and 10 show respectively the time evolution of group cohesion and polarization for the MOST INFLUENTIAL strategy (Panels AD) and the NEAREST strategy (Panels BE), and for $k = 1$ to 4. Despite the absence of confinement due to the wall, all the strategies except the one that consists in interacting only with the nearest neighbor ($k = 1$) allow the group to remain cohesive and polarized for more than 2.5 hours ($\approx 10^4$ kicks) in numerical simulations (see Figs 9A, 9B, 9C, 10A and 10B). When agents only interact with their most influential neighbor, the group is highly cohesive ($C \approx 0.1$ m, Fig 9A), but less than in the arena ($C \approx 0.07$ m, Fig 4C). However, the polarization is higher when the group swim in an unbounded domain (mean of $P \approx 0.93$, Fig 10A) in comparison to the arena (mean of $P \approx 0.78$, Fig 5C). Therefore, the confinement due to the arena reinforces the group cohesion and weakens the group polarization, which still remains at a high level for the MOST INFLUENTIAL strategy.

However, when agents only interact with their first nearest neighbor, the group disintegrates very quickly and then diffuses, with $C^2(t)$ growing linearly with the time $t$ (Fig 9C), and $P(t)$ oscillating around 0.6 (Fig 10B). Compact groups are recovered for the NEAREST strategy with $k = 2$, 3, but the MOST INFLUENTIAL strategy systematically leads to more cohesive and more polarized groups (Fig 9A and 9B).

In order to better understand to what extent the group cohesion depends on the interaction strategy and/or on the long-range nature of the attraction [14], we have also simulated the model by truncating the attraction interaction between two agents $i$ and $j$ when their distance $d_{ij}$ is greater than a cut-off distance $d_{cut}$: $F_{Att}(d_{ij}) = 0$, if $d_{ij} > d_{cut}$, where $F_{Att}$ is defined in Eq (10) of the Materials and Methods section. When $d_{cut}$ decreases below some critical value $d^*_{cut}$, we expect that the group will break and that the agents will ultimately freely diffuse, illustrating the importance of the range of the attraction interaction to ensure the cohesion of the group (see Figs 9D, 9E, 10D and 10E).

For the MOST INFLUENTIAL strategy with $k = 1$, the group remains highly cohesive (Fig 9D) and highly polarized (Fig 10D) for $d_{cut} > d^*_{cut} \approx 0.9$ m. For $k = 2$, 3, and 4, $d^*_{cut}$ is found to be slightly smaller than for $k = 1$ ($d^*_{cut} \approx 0.8$ m; Fig 9D). For the NEAREST strategy with $k = 2$ (the group is never cohesive for $k = 1$, even for $d_{cut} = \infty$; see above), we find $d^*_{cut} \approx 3.5$ m (Fig 9E), much higher than for $k = 1$ in the MOST INFLUENTIAL strategy. Here, we clearly see that even at a smaller $k$, the MOST INFLUENTIAL strategy is much more effective than the NEAREST

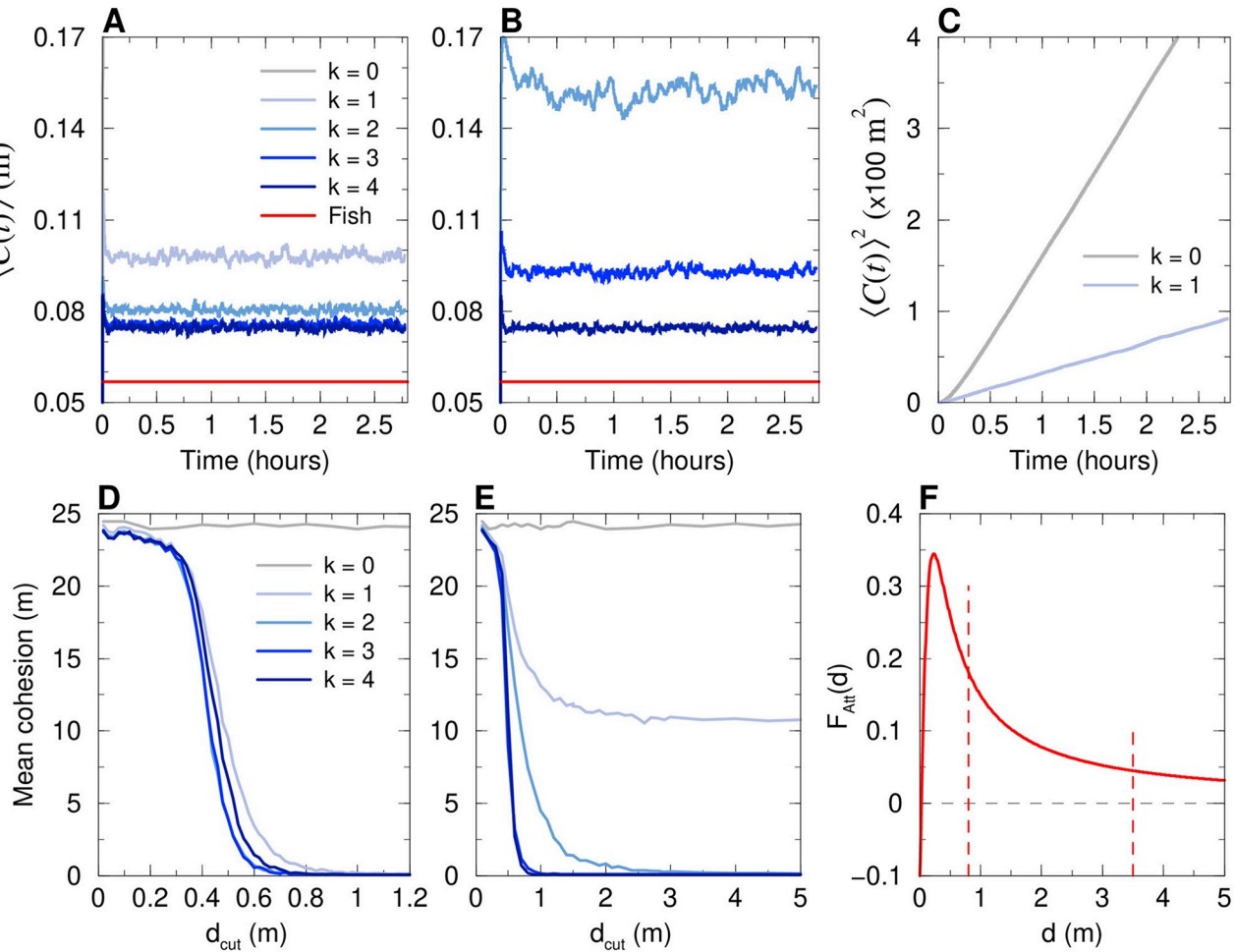

**Fig 9. Average cohesion of a group of 5 agents swimming in an unbounded domain.** Model simulations for the MOST INFLUENTIAL strategy (AD) and the NEAREST strategy (BCE), for $k = 1, \ldots, 4$ (blue lines), together with the case with no interaction ($k = 0$, gray lines) and the mean cohesion for fish experiments (red lines in AB). For $k = 0$, cohesion is lost immediately, so that the gray line is not visible on the scale of panels AB. (C): Squared mean cohesion in the diffusive cases for $k = 1$ nearest neighbor and $k = 0$. (ABC): Average over 1000 runs with 10000 kicks (2.7 hours) per run. (DE): Mean cohesion averaged over the last 10% of the 1000 runs for different values of the cut-off distance $d_{cut}$ for the two strategies: (D) MOST INFLUENTIAL, and (E) NEAREST. Panel (F): We plot the attraction function $F_{Att}$ (see Eq 10), showing the critical values $d_{cut}$ above which cohesion is preserved (vertical dashed lines): $d_{cut}$ 0:8 m when the interacting neighbors are the $k = 1, 2$ or 3 most influential ones, the $k = 3$ nearest ones, or all the neighbors ($k = 4$); $d_{cut}$ 3:5 m when interacting with the $k = 2$ nearest neighbors ($d_{cut}$ does not exist when interacting only with the nearest neighbor).

strategy in ensuring the cohesion of the group, for finite-range attraction cut-off at $d_{cut}$. For $k = 3$, the NEAREST strategy leads to a critical cut-off $d^*_{cut} \approx 0.9$, of the same order as for the MOST INFLUENTIAL strategy (for $k = 3$, the involved neighbors are often the same for both strategies; see above).

In conclusion, for groups of 5 agents in an unbounded domain, we have shown that the MOST INFLUENTIAL strategy leads to a highly cohesive and polarized group for all $k = 1, 2, 3$, provided the range of the attraction is not too small ($d_{cut} > 0.8$ m). For the NEAREST strategy, the group is never cohesive for $k = 1$, and a much larger range of the attraction ($d_{cut} > 3.5$ m) is required to ensure the cohesion of the group for $k = 2$.

**Collective motion of larger groups in an unbounded domain.** For agents moving in an unbounded domain, we have simulated the model with the MOST INFLUENTIAL strategy with $k = 1$, for groups of $N = 6$ to 70 individuals starting initially in a compact configuration (see Fig 10C). The group remains highly cohesive for all sizes (up to $N = 70$), with a group cohesion of

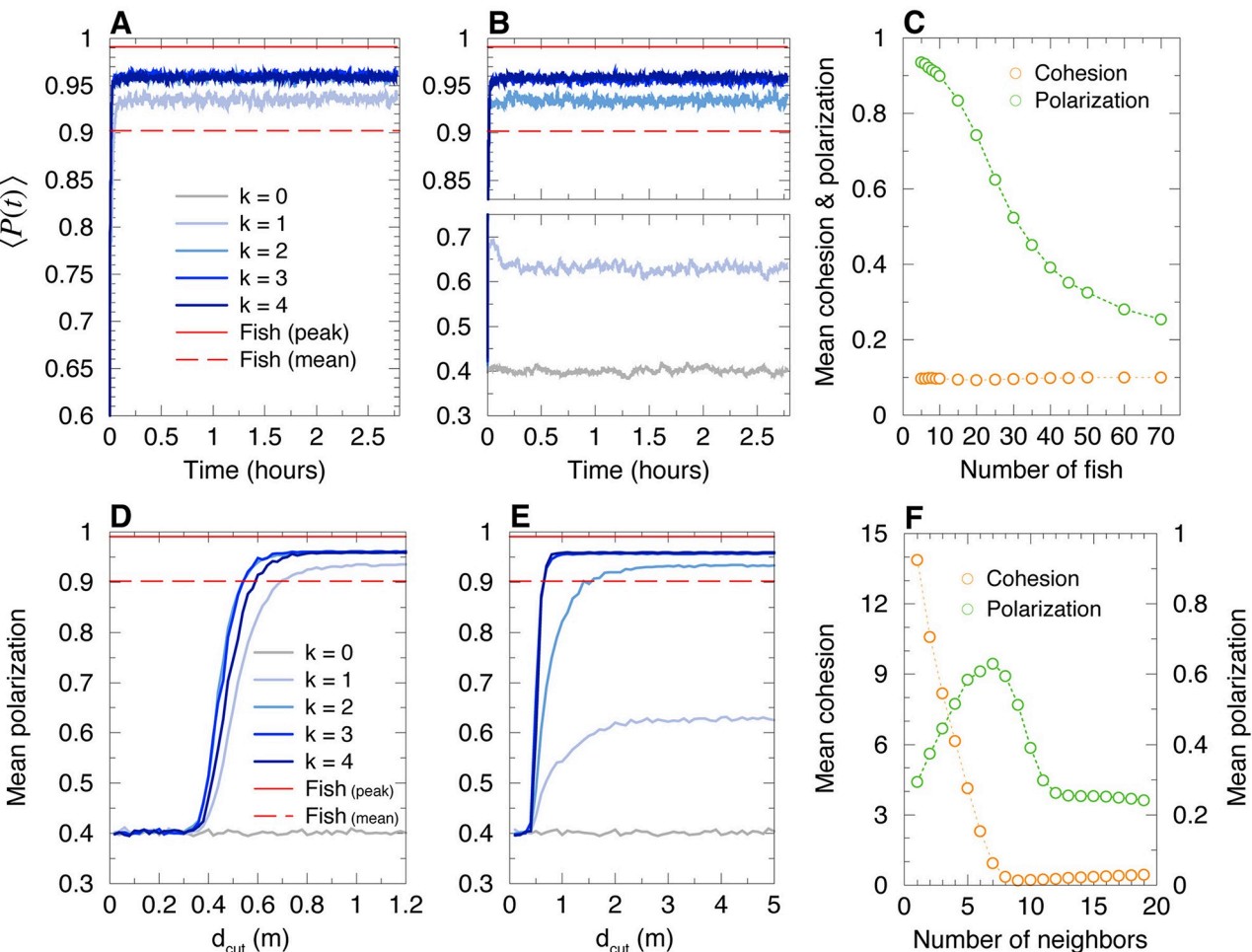

**Fig 10. Average polarization of groups of 5 agents, and mean cohesion and polarization in larger groups ($N$ = 5, …, 70), when agents are swimming in an unbounded domain.** For $N$ = 5, model simulations for the MOST INFLUENTIAL strategy (AD) and the NEAREST strategy (BE), for $k$ = 1, …, 4 (blue lines), together with the case with no interaction ($k$ = 0, gray lines) and the mean polarization for fish experiments (red lines). Panel (C): Mean cohesion and polarization in large groups ($N$ = 5, …, 70) for the MOST INFLUENTIAL strategy ($k$ = 1). Panel (F): Mean cohesion and polarization in a group of size $N$ = 20 as a function of the number $k$ of nearest neighbors with which focal individuals interact. The minimum of the cohesion is reached at $k$ = 9, and the maximum of the polarization at $k$ = 7.

order $C \sim 0.1$ m. The polarization remains high ($P > 0.7$) in groups of size $N \le 20$, and decreases as the group size increases. This suggests a smooth cross-over between a schooling phase up to moderate group sizes $N \sim 20$, and a more disordered swarming phase for larger $N$. In fact, for the largest values of $N$ investigated, schooling periods are also observed, alternating with periods of collective milling, resulting de facto in a reduced polarization of the group. The occurrence of the swarming, schooling, and milling phases as a function of the model parameters (group size $N$, strategy to select the interacting neighbors, intensity and range of the attraction/alignment interactions...) will be studied in a future work, as it has been previously done for the species *Kuhlia mugil* [32] (a species displaying a smooth swimming mode, instead of a burst-and-coast swimming mode).

When agents only interact with their nearest neighbor, groups larger than $N$ = 5 disperse immediately and a larger number of neighbors $k$ must be taken into account to preserve some degree of cohesion. We have also simulated larger groups ($N$ = 6, . . ., 26; $N$ even) with $k$ = 1 to

$N − 1$ for the NEAREST strategy. The results of S7 Fig (and Fig 10F, in the particular case $N = 20$) show that each agent must interact at least with $k \sim N/2$ nearest neighbors in order to obtain a degree of cohesion similar to the one observed for the MOST INFLUENTIAL strategy with $k = 1$. Moreover, once $k > N/2$, groups become less cohesive as the number of nearest neighbors taken into account by agents increases. In fact, for $N > 6$ and whatever the value of $k$, the NEAREST strategy always leads to less cohesive groups (S7A Fig) than for the MOST INFLUENTIAL strategy with $k = 1$, for which $C \sim 0.1$ m.

The simulation results also show that for the NEAREST strategy with $k < 7$, the degree of polarization decreases with the group size. Moreover, the polarization reaches a maximum for $k \sim N/2$ until $N \leq 14$. For larger groups, interacting with more than $k = 7$ nearest neighbors reduces the degree of polarization, which becomes smaller as $k$ increases (see S7B Fig and the particular case of $N = 20$ in Fig 10F).

## Collective behavior in robotics experiments

We now present the results of a series of experiments with $N = 5$ robots exploiting the three interaction strategies considered in the fish model. The robots are programmed to reproduce the model behavior (Eqs (4)–(15) in Materials and methods), with model parameters adapted to the different spatial and temporal scales of the robotic experimental setup (see Table 3). In addition, robots operate a control procedure designed to resolve collisions with the wall, and most importantly, with other robots (see Materials and methods). Indeed, contrary to point particle agents in the fish model or to real fish swimming in shallow water (a truly 3D environment), robots moving on a strictly 2D setup cannot physically cross each other. The robots hence combine a behavioral model and an engineering-minded control system to deal with real-world physical constraints. Our robotic platform provides a concrete implementation of these two elements and understanding their interplay and their combined impact on the collective behavior of robots is certainly one of the main motivation of the experiments presented here.

Panels (DEF) of Figs 4–8 show the results of the robotic experiments performed in the same conditions as those studied with the model, including the case where robots do not interact with each other ($k = 0$) and the case where each robot interacts with all its neighbors ($k = 4$). However, the robotic experiment for the case $k = 3$ for the MOST INFLUENTIAL strategy was not

**Table 3. Values and units of the parameters for model simulations and robot experiments.**

| Parameter | Symbol | Model | Robots |
|---|---|---|---|
| Intensity of heading random fluctuations | $\gamma_R$ | 0.45 | 0.1 |
| Fluctuations reduction factor when close to wall | $\alpha$ | 0.67 | 1 |
| Intensity of wall repulsion | $\gamma_w$ | 0.15 | 0.79 |
| Range of wall repulsion (cm) | $l_w$ | 6 | 11 |
| Intensity of attraction/repulsion | $\gamma_{Att}$ | 0.12 | 0.18 |
| Range of attraction between individuals (cm) | $l_{Att}$ | 20 | 37 |
| Distance of balance of attraction/repulsion (cm) | $d_{Att}$ | 3 | 18 |
| Intensity of alignment | $\gamma_{Ali}$ | 0.09 | 0.04 |
| Range of alignment between individuals (cm) | $l_{Ali}$ | 20 | 37 |
| Distance of alignment (cm) | $d_{Ali}$ | 6 | 5 |
| Average duration between successive kicks (s) | $\tau$ | 0.5 | 1.3 |
| Mean length between two successive kicks (cm) | $l$ | 7 | 7.4 |
| Typical individual velocity in active period (cm/s) | $v_0$ | 14 | 3.75 |
| Relaxation time (s) | $\tau_0$ | 0.8 | 0.9 |

performed. Counter-milling in robots is illustrated in S6 Fig, and the density maps of cohesion and polarization are shown in S3 and S4 Figs. The robotic platform and the monitoring of a group of 5 robots in motion are shown in S2 Video.

Despite the fact that the spatial and temporal scales of the robotic platform have been scaled at best to correspond to that of the fish experiments (in particular, $4 \times 4$ cm square robots in an arena of radius $R = 42$ cm vs elongated fish of typical length 3 cm swimming in a tank of radius $R = 25$ cm), the border and other robots have a stronger effect on a focal robot at short distance. Indeed, as explained above, the collision avoidance protocol (see Materials and methods) induces effective interactions between the robots that have a longer range than the interactions between fish. In addition, the square shape of the robot also makes them effectively bigger than if they were elongated like fish. Hence, the rescaling of distances as measured in robot experiments is necessary to be able to compare the different spatial distributions in fish and robot experiments, although it does not affect polarization, counter-milling, or angular distributions. As a result, we found a much smaller scaling factor than in model simulations: $\lambda_R = 0.35$. Note that once the optimal scaling factor is determined, it is kept fixed in all considered situations (strategy to select the interacting neighbors and their number $k$). From now, all distances in the robot experiments mentioned in this section are hence expressed after rescaling to be comparable to corresponding distances in the fish experiments.

When $k = 0$, robots move independently from each other when they are sufficiently far from each other, and tend to remain dispersed along the border of the arena (S5 Video). The group cohesion is weak (cohesion peaked at $C \sim 12$ cm; Fig 4D, 4E and 4F), and the distance of the barycenter to the wall is large ($r_w^B \sim 12$ cm; Fig 6D, 6E and 6F). Robots are relatively more cohesive and closer to the wall compared to the fish model for $k = 0$ due to volume exclusion effects (two colliding robots can end up going in the same direction as a result of the control procedure) and because the confining effects of the border of the arena are stronger in robots than in agents (see also S3 and S6 Figs). Robots are not polarized, as already observed in the fish model simulations for the same condition $k = 0$ (Panels DEF in Fig 5).

Interacting only with $k = 1$ nearest neighbor does not allow robots to coordinate their motion and move as a coherent group (see S6 Video). Panel (D) of Figs 4–8 (cohesion; polarization; distance to the wall; angle with respect to the wall; counter-milling) show that the results for $k = 1$ are similar to those obtained for $k = 0$, with a marginal improvement of the group cohesion and polarization. On the other hand, when the robots interact with their most influential neighbor (S7 Video), the group is highly cohesive ($C \sim 6.5$ cm; Fig 4F) and highly polarized (large peak at $P = 1$ in Fig 5F). The robots collectively move close to the border ($r_w^B \sim 7$ cm; Fig 6F). Counter-milling is also clearly visible (Fig 8F, S7 Video and S6 Fig). Moreover, for the RANDOM strategy with $k = 1$, the results are somewhat intermediate between those for the NEAREST and MOST INFLUENTIAL strategies, in terms of cohesiveness, polarization, and counter-milling (see Panel E in Figs 4, 5 and 8 respectively, and S8 Video). The similarity of the density maps of cohesion and polarization with those found in fish experiment is the highest for the MOST INFLUENTIAL strategy compared to the other two strategies (S3 and S4 Figs). Overall, and as confirmed by the Hellinger distances listed in Table 2, the MOST INFLUENTIAL strategy with $k = 1$ produces highly cohesive and polarized robot groups leading to a qualitative agreement with fish experiments, whereas the NEAREST strategy does not even lead to any significant group coordination.

Extending the interaction to the $k = 2$ nearest neighbors reinforces group coordination (S9 Video): groups are more cohesive (the peak in the PDF of $C$ decreases from around 10 cm for $k = 1$, to 7 cm), and simultaneously more polarized (S3 Fig). However, the polarization remains weak compared to fish experiments, and even compared to the MOST INFLUENTIAL

strategy for $k = 1$: the PDF of $P$ has a wide region of high values centered in $P \approx 0.85$ and is not peaked at $P = 1$ (Fig 5D). The high peak at $P = 0.6$ reveals that situations in which groups of 4 robots move in the same direction while the fifth robot moves in the opposite direction are quite frequent. Wide groups ($C > 8$ cm, Fig 4D) moving far from the border ($r_w^B > 9$ cm, Fig 6D) are still frequent, and counter-milling is still barely visible (S6 Fig). On the other hand, interacting with the two most influential neighbors definitively produces patterns that are similar to those observed in fish experiments, especially if we consider the polarization, where the peak at $P = 1$ clearly narrows and doubles its height (Fig 5F and S10 Video), although the improvement with respect to the MOST INFLUENTIAL strategy with $k = 1$ is small, or even negligible, if we consider the counter-milling index (Fig 8F). Again, the RANDOM strategy with $k = 2$ leads to an overall much better agreement with fish experiments than the NEAREST strategy with $k = 2$ (see Hellinger distances between PDF in Table 2). Except for the weaker polarization, the results for the RANDOM strategy are similar to the ones obtained for the MOST INFLUENTIAL strategy with $k = 2$ (see Table 2 and S11 Video).

For $k = 3$, the results for the NEAREST strategy (see S12 Video) improve drastically and are in comparable agreement with fish experiments as the results for the RANDOM strategy (S13 Video), and on par with those for the MOST INFLUENTIAL strategy for $k = 1, 2$ (see Table 2). For the NEAREST and RANDOM strategies (sharing 2, and often 3, common neighbors for $k = 3$), groups are highly cohesive (Fig 4D and 4E) and polarized (Fig 5D and 5E), with a narrower PDF of $C$ than in fish experiments, pointing to the robot groups having less internal fluctuations than fish groups. Accordingly, the PDF of $r_w^B$ (Fig 6D and 6E) is peaked at the same value as in fish experiments, $r_w^B \approx 5.5$ cm, but is again narrower, with much less weight at distances $r_w^B > 8$ cm. The PDF of $\theta_w^B$ (Fig 6D and 6E) is in good agreement with fish experiments, and counter-milling is clearly obtained (S6 Fig). When robots interact with $k = 4$ neighbors (S14 Video), the results are very similar to the case $k = 3$ within the non negligible statistical fluctuations due to our shorter robot experiments compared to the fish experiments and fish model simulations.

In conclusion, many of the results of the robotic experiments are qualitatively similar to those found in the simulations of the model, despite the robots being submitted to real-world physical constraints. Yet, for robots, the MOST INFLUENTIAL strategy with $k = 1$ is found to lead to cohesive and polarized groups (like in the model), while the NEAREST strategy with $k = 1$ does not lead to any significant group coordination (weaker coordination for the model in a confining domain, but no cohesive groups in an unbounded domain).

## Discussion

Collective motion involving the coherent movements of groups of individuals is primarily a coordination problem. Each individual within a group must precisely adjust its behavior to that of its neighbors in order to produce coordinated motion. Determining how these relevant neighbors are selected at the individual scale is therefore a key element to understand the coordination mechanisms in moving animal groups. Previous experimental works on fish and birds have identified interacting neighbors using short-term directional correlations [17, 33] or anisotropy of the position of the nearest neighbors [21]. In a starling flocks (*Sturnus vulgaris*), each bird coordinates its motion with a finite number of closest neighbors (typically seven), irrespective of their distance [21]. However, in fish schools, experimental studies suggest that each individual only interacts with a smaller number of influential neighbors. For instance, in the mosquitofish (*Gambusia holbrooki*), each fish mostly interacts with a single nearest neighbor [34]. In the rummy nose tetra (*Hemigrammus rhodostomus*) during collective U-turns [28, 35], the analysis of directional correlations between fish suggests that each fish

mainly reacts to one or two neighbors at a time [28]. These results are in line with theoretical works that have suggested that, instead of averaging the contributions of a large number of neighbors, as suggested by many models [18–20, 23, 36, 37], individuals could pay attention to only a small number of neighbors [25–28, 38]. This mechanism would overcome the natural cognitive limitation of the amount of information that each individual can handle [39].

Here, we addressed this question in groups of five *H. rhodostomus* swimming in a circular tank. This species of fish is of particular interest because of its tendency to form highly polarized groups and its burst-and-coast swimming mode [14], which allows us to consider that each fish adjusts its heading direction at the onset of each bursting phase, that is labeled as a "kick". Just before these brief accelerations, a fish integrates and filters the information coming from its environment and picks its resulting new heading.

In our experiments, groups of five fish remain highly cohesive, almost perfectly polarized, and swim along and close to the wall of the tank, keeping the same direction of rotation for very long periods [35]. Fish groups also display a remarkable counter-milling collective behavior where individual fish rotate around the group barycenter in the opposite direction to that of the group in the tank, so that individuals alternate their positions at the front of the group.

Based on a previous work in which we have reconstructed and modeled the form of the interactions of *H. rhodostomus* fish swimming in pairs [14], we analyzed three strategies for combining the pairwise interactions between a focal fish and a number $k = 1$ to 3 of its neighbors by means of a computational model and a robotic platform. In the NEAREST strategy, neighbors are selected according to their distance to the focal individual. In the RANDOM strategy, neighbors are randomly chosen, and in the MOST INFLUENTIAL strategy, neighbors are selected according to the intensity of their contribution to the heading variation of the focal individual. The impact of these strategies on the resulting collective behavior was then measured and analyzed by means of five quantities: group cohesion, group polarization, distance and relative orientation of the barycenter with respect to the border of the tank, and counter-milling index.

Our results suggest that when individuals (agents or robots) interact with a minimal number of neighbors, namely two, a group of individuals is able to reproduce the main characteristics of the collective movements observed in the fish experiments.

In the simulations of the model for $N = 5$, when the agents are interacting with a single neighbor, this immediately leads to the formation of groups. Whatever the strategy used to select a neighbor, the quantities used to quantify group behavior show that the exchange of information with a single neighbor leads agents to get closer to each other, at least temporarily for the NEAREST strategy. However, whatever the strategy considered, cohesion, polarization, and counter-milling are still weak compared to fish experiments, although the MOST INFLUENTIAL strategy convincingly leads to the best group coordination for $k = 1$.

The simulations of the model in an unbounded domain show that group cohesion is maintained over long periods of time when agents only interact with their most influential neighbor, provided the attraction range is above a critical threshold distance. However, when agents only interact with their nearest neighbor, this systematically leads to the diffusive dispersion of the group. For groups of size up to $N = 70$, interacting with the most influential neighbor leads to compact groups, while one needs to consider typically at least $\sim N/2$ nearest neighbors to achieve the same result for the NEAREST strategy. Therefore, the cohesion of the group observed in the arena is not a merely consequence of the confinement of the agents, but mainly results from the higher quality of the information provided by the influential neighbors in comparison to the one provided by the nearest neighbors.

Then, when agents acquire more information about their environment ($k = 2$), all the interaction strategies implemented in the model give rise to collective behaviors that are in qualitative agreement with those observed in the experiments with fish, and a quantitative agreement

is even reached for some quantities characterizing group behavior (see Table 1). When agents collect even more information about their environment (*i.e.*, when they pay attention to $k = 3$ neighbors), the agreement with fish experiments is not improved if the neighbors are chosen according to their influence. However, groups become more cohesive and polarized when the agents interact with their nearest neighbors. Yet, for $k = 3$, the three strategies lead to comparable results, which is consistent with the facts that two strategies have necessarily at least two common neighbors for groups of five individuals. Note that for $k = 2$ and $k = 3$, and for all three strategies, the intensity of the counter-milling is larger in the model than in fish experiments, suggesting that the internal structure of real fish groups is more rigid than predicted by the model.

In summary, the simulation results clearly indicate that group behaviors similar to those observed in fish experiments can be reproduced by our model, provided that individuals interact with at least two of their neighbors at each decision time and no clear gain is obtained when agents interact with a third additional neighbor. When only one interacting neighbor is considered, the MOST INFLUENTIAL strategy leads to the best group coordination, which even survives when the group moves in an unbounded domain.

By implementing the behavioral fish model and the same local interaction strategies in our robotic platform, we also investigated the impact of the physical constraints and the collision avoidance protocol based on speed control on the group behavior. The MOST INFLUENTIAL strategy is much more efficient than the two other strategies to ensure group cohesion and polarization (see Table 2). Remarkably, and as already observed in the model simulations, even when robots only interact with their most influential neighbor, the group remains highly cohesive and polarized, and close to the border. By contrast, when robots only interact with their nearest neighbor, they are not able to exhibit any kind of coordinated behavior. Everything happens as if pairwise interactions between robots were screened by the effect induced by the collision avoidance protocol: the distributions of the group cohesion, the polarization, and the distance of the barycenter of the group to the border of the tank are almost identical to those obtained with the null model, in which no interaction exists between robots except for collision avoidance. When robots interact with two neighbors, the agreement with the results of fish experiments is improved, but it is only when robots interact with three nearest neighbors that the NEAREST strategy produces highly cohesive and polarized groups.

Overall, and even more convincingly than in the case of the fish model, the MOST INFLUENTIAL strategy leads to the best overall agreement with fish experiments for $k = 1$ and $k = 2$, even producing strongly coordinated groups for $k = 1$. Compared to the case of the fish model, the NEAREST strategy does not lead to any significant group coordination for $k = 1$, and only to moderately cohesive and polarized groups for $k = 2$, yet being even less efficient than the RANDOM strategy. The robot collision avoidance protocol induces a strong effective repulsion between close neighbors, which screens the behavioral interactions for the strategy based on these nearest neighbors.

Note that implementing the $k$-MOST INFLUENTIAL strategy in a computational model for larger groups of agents is not more computationally challenging than the implementation of the more common $k$-NEAREST strategy, and is even less demanding than the consideration of the first layer of neighbors in a Voronoi construction used in many phenomenological flocking models [21, 22, 32]. For very large groups ($N > 10000$), rarely considered in the context of fish models, the implementation of the $k$-MOST INFLUENTIAL and $k$-NEAREST strategies could also be optimized by exploiting grid algorithms commonly used in computational physics and astrophysics.

However, beyond its purely computational complexity, the possible biological relevance of the MOST INFLUENTIAL strategy (with small $k$) for fish and potentially other animals is certainly

an important question. In vertebrates, and in particular in fish, the midbrain and forebrain networks are carrying out computation in parallel to process the visual information and select the most salient stimuli that are the focus of attention. The midbrain network continuously monitors the environment for behaviorally relevant stimuli [40]. This is a primary site where the information about the neighbors is filtered for cognitive decision. Then, the forebrain network selects those stimuli on which the fish focuses its attention. The interaction strategies that we have investigated in this work correspond to different ways for an individual to focus its attention on the stimuli (*i.e.*, its relevant neighbors). In the context of fish schools, individuals filter the information from their environment and thus limit their attention to a small set of their most salient neighbors [25–27], hence giving priority to the few neighbors to be avoided (by moving away or by aligning their headings) or the ones to be followed. These few neighbors requiring an immediate action from the focal fish should, by definition, trigger a larger response than other neighbors, hence making the notion of most influential neighbors quite natural. Our results show that each fish interacts with typically two neighbors that are the most salient, a process which reduces the amount of information that needs attention and which hence permits to avoid any cognitive overload.

In conclusion, each individual must acquire a minimal amount of information about the behavior of its neighbors for coordination to emerge at the group level, thus allowing fish to avoid information overload when they move in large groups [39].

## Materials and methods

### Fish experiments

**Ethics statement.**   Our fish experiments have been approved by the Ethics Committee for Animal Experimentation of the Toulouse Research Federation in Biology N˚ 1 and comply with the European legislation for animal welfare.

**Study species.**   Rummy-nose tetras (*Hemigrammus rhodostomus*) were purchased from Amazonie Labège in Toulouse, France. Fish were kept in 150 l aquariums on a 12:12 hour, dark:light photoperiod, at 25.2˚C (±0.7˚C) and were fed *ad libitum* with fish flakes. The average body length of the fish used in these experiments is 31 mm (±2.5 mm).

**Experimental setup.**   We used a rectangular experimental tank of size $120 \times 120$ cm, made of glass, supported by a structure of metal beam 20 cm high. A plywood plate was interposed between the mesh and the basin to dampen the forces exerted on the glass basin by its own weight and water. This structure also enables the attenuation of vibrations. The setup was placed in a chamber made by four opaque white curtains surrounded by four LED light panels to provide an isotropic lighting. A circular tank of radius $R = 250$ mm was set inside the experimental tank filled with 7 cm of water of controlled quality (50% of water purified by reverse osmosis and 50% of water treated by activated carbon) heated at 24.9˚C (±0.8˚C). Reflection of light due to the bottom of the experimental tank is avoided thanks to a white PVC layer.

Each trial started by placing groups of $N = 5$ fish randomly sampled from the breeding tank into the circular tank. Fish were let for 10 minutes to habituate before the start of the trial. A trial then consisted of one hour of fish freely swimming in the circular tank with experimenters out of the room. Fish trajectories were recorded by a Sony HandyCam HD camera filming from above the setup at 25 Hz (25 frames per second) in HDTV resolution ($1920 \times 1080$p). We performed 11 trials with groups of $N = 5$ fish, and for each trial, we used different fish taken from the breeding tank.

## Robotic platform

**Robots.** We used a robotic platform composed by small compact mobile robots that we named "Cuboids", a name chosen in reference to the first realistic computer program that simulated the flocking behavior in birds and the schooling behavior in fish, called "Boids", developed in 1986 by Craig Reynolds [41]. The Cuboids robots were specifically designed by us for this experiment.

Cuboids have a square basis of 40 × 40 mm, they are 60 mm high and weigh 50 g (Fig 11). We now describe the elements of a Cuboid (numbers between parentheses refer to labels in Fig 11). Each robot is equipped with two differential wheels (7) driven by small DC motors (13). The small belts (9) connect wheels to the DC motors, which can drive the robot with a maximum speed of 50 mm/s. The two wheels are mounted on a central axis (6). An IEEE 802.11n/WIFI module (8) with a range of approximately 200 m is used for communication network between robot and a wireless router. A Li-Poly rechargeable battery (15) provided energy for about 6 hours in our experimental conditions. In addition, a coil (12) located under the robot, can be used to charge the robot wirelessly while it is working. The charging circuit is located on the side board (11). The robot bottom hosts a 32-bit, 168 MHz ARM microprocessor STM32F4 (14), which can provide multi control loops with the time duration up to 2 ms. Besides, another 8-bit microcontroller PIC18F25k22 is mounted on the top sensor board (1), which controls a LCD screen (16) to display information and a 3-colors LED (17). The microprocessor communicates with the microcontroller by 4 copper bars (4), which can simultaneously provide power and communication bus.

Each Cuboid also has several sensors to measure the relative positions of other robots in its neighborhood and to send and receive messages from these robots. Within a sensing range of about 20 cm, a robot can send messages (infrared signals) by the center IR transmitter (3). There are two IR receivers (2) on both sides of the robots, which can determine the distance of a neighboring robot that transmits the infrared signal. From the two distance values provided

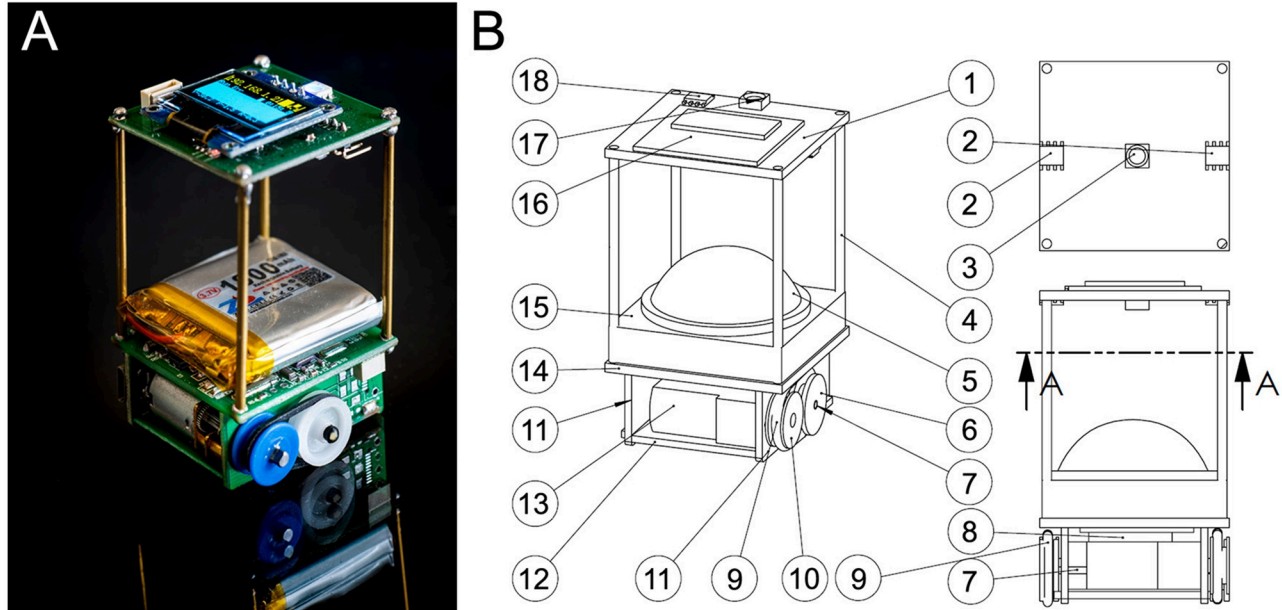

**Fig 11. Cuboid robots.** (A) Photograph of a Cuboid robot. Credits to David Villa ScienceImage/CBI/CNRS, Toulouse, 2018. (B) Design structure of Cuboid robot; A-A represents a cutaway view.

by the IR receivers, the angle with which this neighboring robot is perceived by the focal robot can be calculated by triangulation. Furthermore, the relative position of the neighboring robot to the focal one can be computed by the information of the distance and the angle of perception acquired before. On the other side, the IR signal also carries a short message that includes information on robot ID, orientation angle, speed and states. The heading of a Cuboid is measured by a motion tracking sensor MPU-9250 (18). This device consists of a 3-Axis gyroscope, a 3-Axis accelerometer, and a 3-Axis magnetometer. Hence, the MPU-9250 is a 9-axis Motion Tracking device that also combines a Digital Motion Processor. With its I2C bus connected with PIC18F25K22, the MPU-9250 can directly provide complete 9-axis Motion Fusion output to the microcontroller. These sensing and local communication devices have not been used in the experiments that have been done in a supervised mode.

We tested the model with the robotic platform because there are many physical aspects that have to be considered to assess the robustness of the coordination mechanisms when they are implemented in a physical hardware. These physical aspects include the friction of wheels, the noise of gear box, the blurring of the camera, the wrong identification of the tracker, the delay of the communication, the overload of computation, the blocking of the onboard communication bus, the square shape collision of the robot frame, the mismatch parameters of the interaction model, the impact of the obstacle avoidance protocol, and the non-holonomic constraint of the robot. All these physical aspects can have a large effect on the individual and collective behaviors (especially when robots move in a crowded space) and are difficult to include in a model.

**Experimental platform.** The robotic experimental setup consisted of a circular arena of radius 420 mm resting on a $1 \times 1$ m square flat surface with a camera (Basler piA2400-17gc) mounted on the top (see Fig 12). The setup was placed in a chamber made by 3 opaque wooden boards and 1 white curtain. 2 LED light panels provide a diffused lighting. A circular cardboard wall of radius $R = 420$ mm delimited the border of experimental platform. The floor of the experimental platform was made with a rough wooden board that prevented the reflections of light. A computer is connected to the camera to supervise the actions of the robots in the arena, and to perform the necessary image processing to track each robot and compute in real time its position $(x, y)$ and heading angle $\phi$.

The clock cycle of the imaging process module is 300 ms, a limit imposed by the camera updating speed. A tracking software (Robots ID Tracker), based on the Kalman filter technology, is then used to assign the location data to the right robots on a shorter time scale (every 20 ms). These data are used in real time to control the reaction of each robot in its changing environment, and are also stored in the computer for off-line *a posteriori* trajectory analysis. Thanks to the high precision of our tracking system, we are able to compute in real time and for each robot $i$ the quantities that characterize its instantaneous state with respect to its environment: the distance and relative orientation to the wall $r_w^i$ and $\theta_w^i$, and the distance, relative angular position, and relative orientation with respect to other robots $j$, $d_{ij}$, $\psi_{ij}$ and $\phi_{ij}$, respectively (Fig 2). All this information is used to compute the output of the interactions of a robot with its local environment by means of an Object-Oriented Programming software developed by us. The robot behavior is driven by the mathematical fish model, which combines the interactions with the obstacles and with the other robots, and generates the control signals dispatched in a distributed way to each individual robot through a WIFI communication router (HUAWEI WS831).

Although the robot has its own sensors to ensure it autonomous control and movements, in this work, we used a remote-control mode. This is because our goal was to compare the performances between the software simulation and the robot experiment with the same

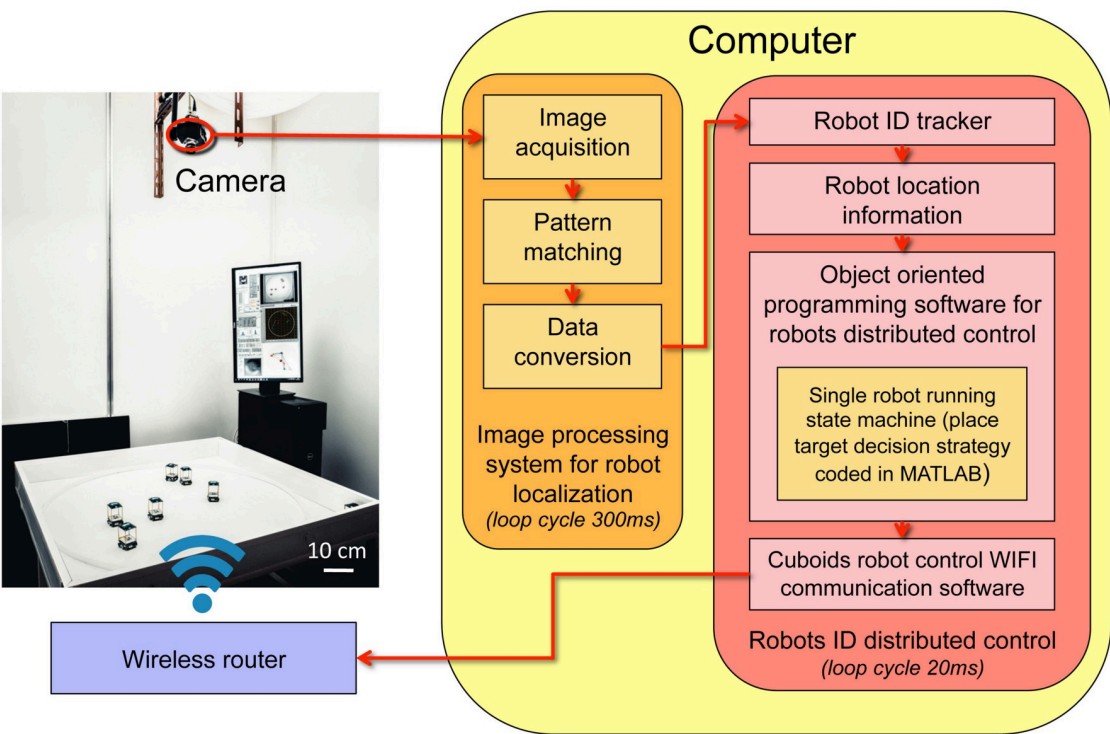

**Fig 12. Structure of Cuboids platform.** Two main parts: the physical hardware and the control software. The hardware consists of a square platform. A camera mounted on the top of it monitors the movements of Cuboids robots, which are controlled in a distributed way by a wireless router. The software processes the image acquired by the camera, then computes the command of actions to be performed by each robot, and finally sends the control signals to the robots via the WIFI channel. Then, all the robots execute their commands at the same time to perform the collective motion. The WIFI broadcasting is one-way communication for sending the command to the robots every 20 ms. In this setup, no information acquired by robots sensors is sent back to the computer though the WIFI channel. Credits to David Villa ScienceImage/CBI/CNRS, Toulouse, 2018.

computational model and the same local information input (see the Hardware In Loop simulation in Fig 13; [42]).

Fig 13 (red and blue boxes) shows the "Hardware In Loop" (HIL) simulation used to control the Cuboids robots. The HIL simulation integrates the robots hardware into the distributed control loops of the platform computer software. As such, it differs from a traditional software simulation, being a semi-real one. Compared with pure theoretical simulations "in silico" (*i.e.*, the software simulation box in Fig 13), the HIL simulation integrates both the hardware constraints (*i.e.*, the mechanical constraints of the robots, the time delay of the control loop which includes the shooting by the camera, the time of calculation and sending orders by the WIFI router), and those that result from the movement of the robots in a physical environment, in particular the need to avoid collisions with obstacles and other robots (see the blue box in Fig 13).

The main difference between the HIL simulation and the software simulation is the real time control of the behavior of each robot, which is achieved by the *Motion Control* and the *Real Time Control* modules (see the red box in Fig 13).

The Motion Control module can produce two kinds of motion patterns: rotating and moving straight. The first motion pattern is *Spot Rotation*, which means that the robot rotates around its center by means of wheels differential driving. The speed control of the two wheels

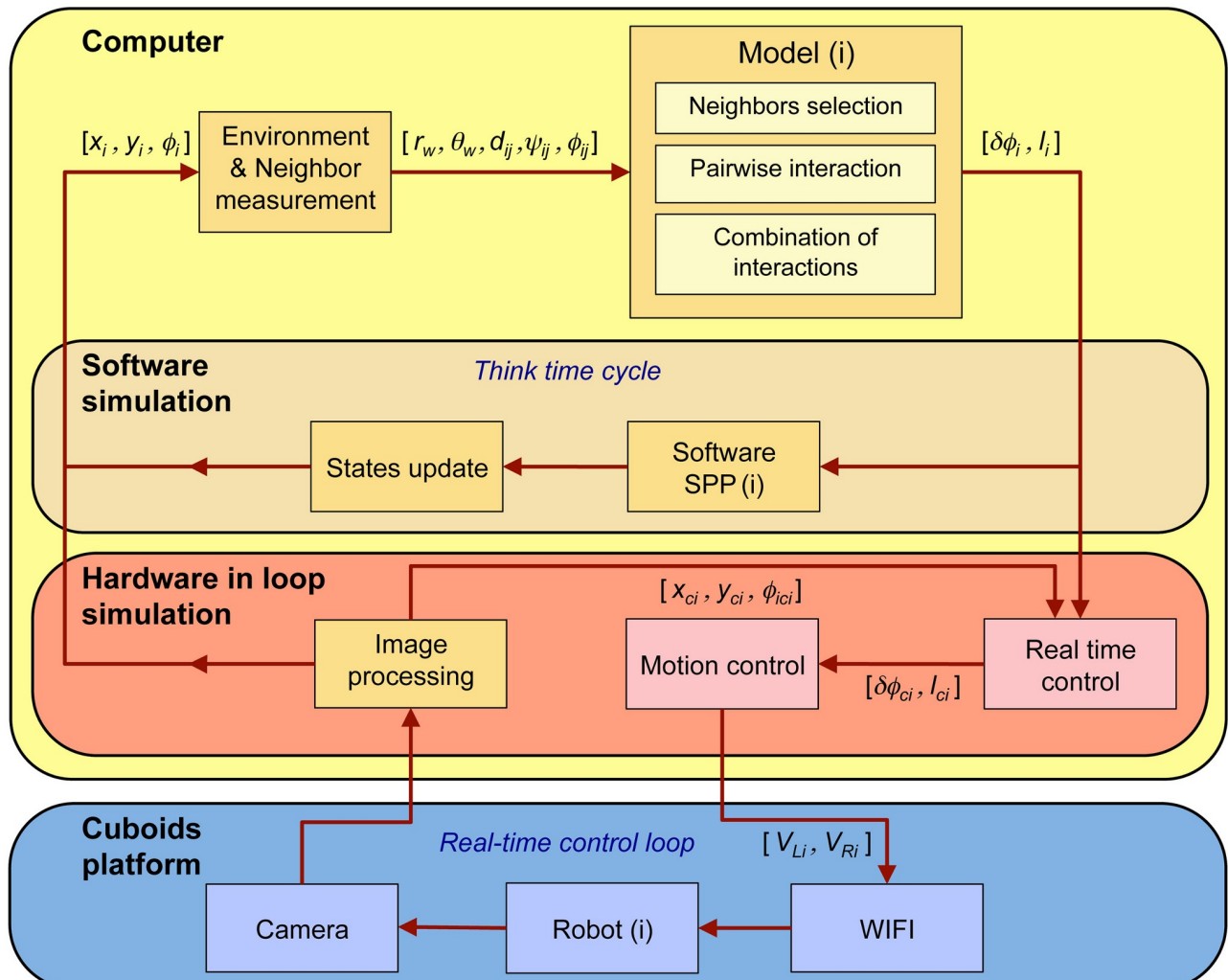

**Fig 13. Software simulation and Hardware in Loop (HIL) simulation (from [42]).** The structure of HIL is an extension of the software simulation, which consists of two extra parts: 1) a computer software (Image Processing, Motion Control, and Real Time Control modules) and 2) a physical hardware (Robot, camera and wireless router). In the software simulation, the Environment & Neighbor Measurement module converts the global position of a robot or a particle $(x_i, y_i, \phi_i)$ in the SPP software into local information $(r_{w,i}, \theta_{w,i})$ and $(d_{ij}, \psi_{ij}, \phi_{ij})$. Then the computational model generates a new kick decision in the form of heading variation and kick length $(\delta\phi_i, l_i)$. This new decision $(\delta\phi_i, l_i)$ is then directly sent to the SPP($i$) software. Once the state has been updated, a new global position is provided by the SPP($i$) software (brown box) or the Hardware in loop simulation (red box). By contrast, the HIL simulation includes hardware, *i.e.*, robots, camera and WIFI router (blue box). Furthermore, each robot $i$ is controlled in real time by three more software modules running in the computer, which are the *Image Processing*, *Motion Control*, and *Real Time Control* modules (red box). The Image Processing module computes the global position of each robot $(x_{c,i}, y_{c,i}, \phi_{c,i})$ from the information provided by the camera in real time. Then, the Real Time Control module converts the model decision $(\delta\phi_i, l_i)$ into a real time decision in the robot $(\delta\phi_{c,i}, l_{c,i})$, which are the heading variation and kick length to perform the decision based on its real time position $(x_{c,i}, y_{c,i}, \phi_{c,i})$. Finally, the Motion Control software generates left and right wheel motors speed control $(V_{L,i}, V_{R,i})$ for each robot to achieve its decision $(\delta\phi_{c,i}, l_{c,i})$. Each robot receives these motor commands by WIFI signals, and performs the corresponding movements that are monitored by the camera.

is described by the following equation:

$$V_{\mathrm{R},i} = -V_{\mathrm{L},i} = p_t\,\delta\phi_{ci},$$

where $V_{\mathrm{R},i}$ and $V_{\mathrm{L},i}$ are the speeds of the right and left wheels of the robot respectively, $p_t$ is a constant factor of proportionality, and $\delta\phi_{ci}$ is the real-time value of the heading variation, which is determined by the Real Time Control module. The second motion pattern that the

robot can display is *Moving Straight*, where the speeds of the left and right wheels are the same:

$$V_{R,i} = V_{L,i} = p_m \, l_{ci},$$

where $p_m$ is a constant factor of proportionality and $l_{ci}$ is the value of the kick length, which is also determined by the Real Time Control module.

The Real Time Control module ensures the safe movement of each robot and helps the robot to rotate and move straight towards a target place. This module first converts the decision of the computational model $(\delta\phi_i(t_{dec}), l_i(t_{dec}))$ at the last decision time $t_{dec}$ into a real-time decision that is then performed by the robot, $(\delta\phi_{ci}(t), l_{ci}(t))$, $t > t_{dec}$.

**Algorithm of the Real Time Control module**

```
Input:
  Computational Model decision: (δφᵢ(t_dec), lᵢ(t_dec)),
  Current position and heading: (x_ci(t), y_ci(t)), φ_ci(t).
Output:
  Real-time decision: (δφ_ci(t_dec), l_ci(t_dec))
  1. δφ_ci(t) = φᵢ(t_dec) + δφᵢ(t_dec) − φ_ci(t),  t > t_dec.
  2. l_ci(t) = lᵢ(t_dec) − √((x_ci(t) − xᵢ(t_dec))² + (y_ci(t) − yᵢ(t_dec))²).
  3. If |δφ_ci(t)| > δφ_Threshold, then
        Do Spot rotation for Motion Control in real-time.
     else
        Do Moving straight for Motion Control in real-time.
  4. If |δφ_ci(t)| < δφ_Threshold and l_ci(t) < l_Threshold, then
        the target is reached;
        Goto Compute state (computational model) for a new decision.
  5. If the path is not free, then
        Do Obstacle Avoidance procedure.
  6. End
```

There are two time-scales in the control for the robots. The long time scale is determined by the time taken in simulating the computational model, which is about 1.3 s. The short time-scale corresponds to the Real Time Control module, which operates at a high frequency with respect to the real time motion of the robot. This module is used to control the navigation of the robot toward the target and the obstacle avoidance (see the table of Algorithm of the Real Time Control module). The time interval of the Real Time Control module is 20 ms for each robot. With such a fast frequency, the communication channel is always busy. To solve this problem, we designed and used a specific protocol to broadcast in one loop the Motion Control command $(V_{R,i}, V_{L,i})$ to each respective robot $i$, based on TCP protocol, thus guaranteeing the speed and the robustness of the communication channel. The average duration of one of these loops (for all robots) in the Real Time Control is about 13 ms (Fig 13), which is less than the fixed time interval of Real Time Control module.

**Implementation of the behavioral model in the robots.** We use the LabVIEW object-oriented programming (OOP) tool to design the distributed control software for the Cuboids robots (Fig 13). It first establishes independent memories for each robot as an agent to store real time information, such as robot ID, location and heading $(x_{ci}(t), y_{ci}(t), \phi_{ci}(t))$ at time $t$, and real time decision $(\delta\phi_{ci}(t), l_{ci}(t))$. We design a state machine control structure to implement the HIL simulation control for each robot. With the new speed control command determined by the *Motion Control* module, the actuators of the robot are controlled wirelessly by the WIFI signals sent by the computer. The robot controls its wheels to move towards the new target place while LED colors display the state of the robot.

Robots use a constant kick length $l_i(t_{dec})$ of around 8 cm, that is, twice the body length of a robot, which corresponds to the mean kick length measured in experiments with five fish.

Using a constant straight step also allows to check if the new target place can be reached or not, in particular, to prevent the case where the agent could be intercepted by another agent, in which case the distance traveled by the agent will be shorter than $l_i(t_{dec})$.

The state machine control structure for an individual robot includes two main states: COMPUTE state and MOVE state; see the flow chart of the robot state machine and the finite state machine diagram in Fig 14 and S8 Fig respectively. The robots are programmed to perform a burst-and-coast movement mimicking the swimming mode of fish. When a robot is in the COMPUTE state at time $t_{dec}$, the computational model determines a new decision ($\delta\phi_i(t_{dec})$), $l_i(t_{dec})$ (see hereafter and [14] for the description of the model; the model parameter for the robots are listed in Table 3). After that, the robot switches to the MOVE state and adjusts its wheels to move towards the decision place in real time thanks to the *Motion Control* and *Real Time Control* modules. Since other robots are moving around asynchronously, the robot must avoid these dynamic obstacles while being in the MOVE state. To prevent collisions between robots, we designed and implemented an obstacle avoidance protocol. When no valid targets can be generated during the COMPUTE state (due to the impediment imposed by nearby robots), the robot generates a valid target place by means of a scanning method and, alternatively, just moves back over a short distance. However, this circumstance rarely occurs in our experiments (except in the absence of behavioral interactions, $k = 0$; see S5 Video).

We describe below the two states and the additional procedures used to avoid collisions with dynamical obstacles.

- COMPUTE State: This state generates a new decision ($\delta\phi_i(t_{dec})$, $l_i(t_{dec})$) for the focal robot by means of the computational model, which is programmed in MATLAB. In this state, the robot takes the information about its local environment ($r_{w,i}$, $\theta_{w,i}$) and selects the neighbors to be taken into account corresponding to the current local interaction strategy. Then, the robot computes the variation of its heading angle ($\delta\phi_i(t_{dec})$) that, combined with the kick length $l_i(t_{dec})$, determines a new target place. The location of the new target is then checked and validated by the OOP software so as to avoid any collision with static obstacles, before the robot switches to the MOVE state (see Fig 14, S8 Fig). While a robot is in the COMPUTE State, the white LED light is turned on.

- MOVE State: In this state the robot evaluates whether its heading angle $\phi_{ci}$ is aligned with the new pace target. If the deviation $\delta\phi_{ci}$ is too large, the robot first rotates towards the target and then moves straight until it reaches the target, thanks to the *Motion Control* module. Then, when the robot successfully reaches the target, it returns to the COMPUTE state to determine a new target. While a robot is in the MOVE State, the green LED light is turned on.

- Obstacle Avoidance Protocol: This procedure is triggered as soon as the target path of the focal robot $i$ crosses the safety zone of another robot $j$. The safety zone is a circular area around a robot of diameter of 80 mm. In this case, the focal robot $i$ first stops and computes whether it can continue moving or not, according to the information it has about the distance $d_{ij}$ and relative angular position $\psi_{ij}$ of the neighboring robot. If the focal robot has the moving priority (determined by a large value of the angle of perception, $|\psi_{ij}| > 90°$, meaning that the robot is a temporary leader [14]), or if the distance is larger than the diameter of the circle of security ($d_{ij} > 80$ mm, meaning that the robot $j$ is far enough), the moving condition is satisfied and the focal robot $i$ successfully switches back into the MOVE state. If not, it repeatedly checks the values $d_{ij}$ and $\psi_{ij}$ until the moving condition is satisfied. If the focal robot cannot go back into the MOVE state within 3 seconds, it toggles to the COMPUTE state to determine a new target.

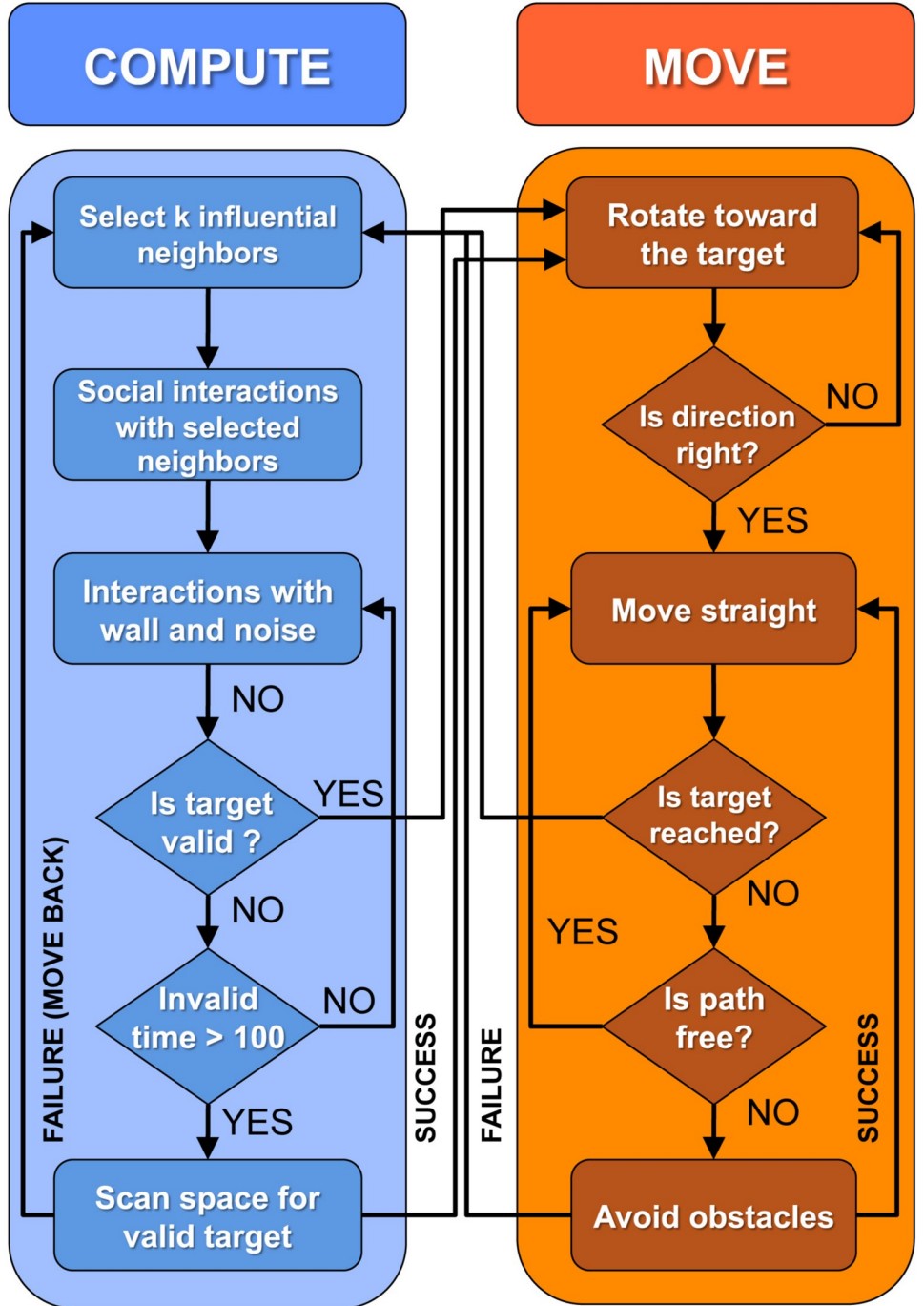

**Fig 14. Flow chart of robot states machine.** At any time a robot can be in one of the two following states: (1) the COMPUTE state for choosing a new target place, and (2) the MOVE state to reach the target place. In the COMPUTE state, the robot first selects influential neighbors, then it computes the pairwise influence of each neighbor, and finally it adds all influences to generate a new target place. Then, this new target place is validated to avoid collisions with the wall or another robot. If a valid target place cannot be found, the robot scans all space around itself for a valid target place. If the scanning method cannot find a valid target, the robot moves back over a distance of 80 mm and starts again the COMPUTE state. When a valid target place has been found, the robot switches into the MOVE state. The robot first rotates toward the target and then, moves straight to it. If another running neighbor blocks the path, the robot uses a procedure to avoid the obstacles.

- No Valid Target Procedure: This procedure is triggered when the robot is in the COMPUTE state and cannot generate a valid target place within 3 seconds. In this situation, the robot scans the local environment from its front to the nearest neighbor located at one of its sides. If there exists a free space for generating a target place, the robot toggles to the MOVE state. If, after scanning, no free space is available for moving, the robot moves back over a predefined distance of 80 mm (approximately two robot body lengths) and then toggles to the COMPUTE state to determine a new target place.

In the robotic experiments, we performed one experiment for each combination of interactions with about 8000 kicks in average for all the 5 robots. The duration time of experiments performed for each condition was the following:

- Interacting with $k$ = 1, 2 and 3 nearest neighbors: 61, 62 and 63 min respectively.

- Interacting with $k$ = 1, 2 and 3 randomly chosen neighbors: 65, 128 and 48 min respectively.

- Interacting with $k$ = 1 and 2 most influential neighbors: 68 and 82 min respectively.

- Interacting with $k$ = 0 and $k$ = 4 neighbors: 150 min in both cases.

## Data extraction and preprocessing

Fish data were extracted from videos recorded during 11 sessions along 11 days in 2013, by means of idTracker software version 2.1 [43], producing 11 data files with the position (in pixels) of each fish in each frame, with a time step of $\Delta t$ = 0.04 s (corresponding to images taken with a frequency of 25 fps). Data were located in a rectangle of size [471.23, 1478.48] × [47.949, 1002.68] containing the circular tank of diameter 50 cm. The conversion factor from pixels to meters is $0.53 \times 10^{-3}$ m/pix. The origin of coordinates $T(0, 0)$ is set to the center of the tank (Fig 1).

We found that trajectory tracking was satisfactorily accurate. However, fish were often misidentified, making impossible the direct use of the data provided by the tracking system. We thus implemented a procedure of identity reassignment that provided us with the proper individual trajectories. In short, the procedure is a sorting algorithm where fish identities are successively reassigned in such a way that the coordinates of each fish at the next time step are the closest ones to the coordinates they had at the previous time. That is, the fish $i$ at time $t$ is assigned the coordinates of fish $j$ at time $t + \Delta t$ that minimize the distance covered by the 5 fish.

Data were then grouped in a single file, counting 1.077.300 times, *i.e.*, almost 12 hours where the position of each fish is known. Then, times where at least one fish freezes were removed. Fish often remain stationary. We considered that a fish is at rest when the distance covered in 60 frames is smaller than 30 pixels, that is, when the mean speed is smaller than 6.6 mm/s during at least 2.4 seconds. We discarded more than half of the data using this procedure (around 5.5 hours of data remaining). We then extracted the continuous sequences lasting at least 20 seconds, obtaining 293 sequences for a total duration of around 3h 10mn.

Fish trajectories were then segmented according to the burst-and-coast typical behavior of this species [14] (see Fig 1C). We used a time window of 0.2 s to find the local minima of the velocity. These points are used to define the onset of a kick event. We detected 60312 kicks, which means that a fish makes in average around 1 kick/s.

In [14], no statistically meaningful left/right asymmetry in the trajectories of single fish ($\sim$ 300000 kicks recorded) or pairs of fish ($\sim$ 200000 kicks recorded) was observed. Hence, for any observed trajectory, the mirror trajectory (that is the same one, but as observed from the

bottom of the tank instead of from the top) would have exactly the same probability to be observed. Assuming the absence of left/right asymmetry for groups of 5 fish (as observed for 1 and 2 fish), leads to the same conclusion. Groups of 5 fish (as well as groups of 5 model fish or 5 robots) rotate clockwise (CW) or counter-clockwise (CCW) around the center of the tank for long periods (collective U-turns in groups of 2-20 fish have been studied in [35]). Therefore, for the much shorter present fish (and especially robots) experiments compared to [14] (60312 recorded kicks, instead of $\sim 500000$), one would observe an artificial asymmetry (groups turning more often CW than CCW, or the opposite) only due to the lack of statistical sampling of the rare collective direction changes. In order to avoid this artificial asymmetry, for each set of 5 trajectories (fish and robots), we have added the mirror set (the trajectories as seen from the bottom of the tank). Again, this procedure is perfectly sound once the absence of left/right asymmetry observed in very long 1- and 2-fish experiments is reasonably assumed to hold in our present 5-fish experiments (the model and its version implemented in robots have obviously no left/right asymmetry, per construction). Note that only the distribution of $\theta_w^B$ is affected by this symmetrization procedure, and not the distributions of group cohesion, polarization, distance to the wall, counter-milling index (the latter being a relative quantity), which are invariant by the mirror symmetry.

To calculate the heading angle of a fish at time $t$, we considered that the direction of motion is well approximated by the velocity vector of the fish at that time $t$. The heading angle $\phi(t)$ of a fish is thus given by the angle that its velocity $\vec{v} = (v_x, v_y)$ makes with the horizontal line, that is,

$$\phi(t) = \text{atan2}(v_y(t), v_x(t)). \qquad (1)$$

Positive angles are measured in counter-clockwise direction and atan2 returns a value in $(-\pi, \pi]$. The components of the velocity are estimated with backward finite differences, *i.e.*, $v_x(t) = (x(t) - x(t - \Delta t))/\Delta t$ and $v_y(t) = (y(t) - y(t - \Delta t))/\Delta t$.

The robot trajectories were extracted with a custom-made tracking software based on Kalman filter and pattern recognition technology [44]. Data were recorded every $\Delta t = 0.04$ s, and trajectories were then subjected to the same treatment.

## Computational model

We use the same model to describe the time evolution of agents in the simulations and to control the decisions of the robots in the experiments, albeit with different parameters to accommodate for the different spatial and temporal scales in the two cases (see Table 3).

*Hemigrammus rhodostomus* displays a "burst-and-coast" swimming behavior characterized by sequences of sudden speed increases called "kicks", each followed by a quasi-passive deceleration and gliding period along a near straight line until the next kick (Fig 1C, S1 and S3 Videos).

In our model, we consider that a fish makes the decision to change its heading and to pick its new kick length and duration exactly at the onset of each kick [14]. The behavior of an agent $i$ is thus described by a sequence of kicking times $t_i^n$ at which the agent $i$ performs its $n$-th kick. An agent selects a new heading depending on the *instantaneous* state of its environment (other fish; obstacles), as perceived exactly at the onset of a new kick, although the results of [28] suggest that the integration of the necessary information by an actual fish can take a few tenths of a second during the previous gliding period. Hence, at each of its kicking times $t_i^n$, the agent $i$ collects the information of its instantaneous relative position and heading with respect to the obstacles and to the other agents, and selects the length and duration of its $n$-th kick, $l_i^n$ and $\tau_i^n$ respectively, and its change of direction, $\delta\phi_i^n$. Each agent has its own sequence

of kicking times, which are not necessarily equally spaced: $t_i^{n+1} - t_i^n \neq t_i^n - t_i^{n-1}$. In addition, the motion of the different agents is asynchronous and their respective kicking times are in general different. As the environment changes from one kick to another (the agent moves with respect to the obstacles, and the other agents move with respect to the agent), the quantities $l_i^n$, $\tau_i^n$, and $\delta\phi_i^n$ are updated at each kicking time of agent $i$, according to the number and identity of the agents taken into account in the evaluation of the effect of social interactions. In the present work, the number of agents taken into account in the social interactions remains constant, while the identity of the neighbors considered to interact with an agent is updated at each kicking time of this focal agent.

The behavior of agent $i$ (fish or robot) is thus described by the following discrete-time decision model:

$$\vec{u}_i^{\,n+1} = \vec{u}_i^{\,n} + l_i^n\,\vec{e}\left(\phi_i^{n+1}\right), \tag{2}$$

$$\phi_i^{n+1} = \phi_i^n + \delta\phi_i^n, \tag{3}$$

where $\vec{u}_i^{\,n+1}$ and $\phi_i^{n+1}$ are the vector position and the heading of agent $i$ at the end of its $n$-th kick, and $\vec{e}\left(\phi_i^{n+1}\right)$ is the unitary vector pointing in the direction of angle $\phi_i^{n+1}$. At the end of the $n$-th kick of agent $i$, the time is $t_i^{n+1} = t_i^n + \tau_i^n$, which is the next kicking time of agent $i$. Note that one or more agents can perform one or more kicks between two successive kicks of agent $i$. In that case, the kicking agent collects the information about agent $i$ (relative position and heading) to perform its own kick, while agent $i$ is simply in the gliding phase following its last kick.

The kick length $l_i^n$ is sampled at each kicking time of agent $i$ from the bell-shaped distribution of kick lengths obtained in our experiments of fish swimming in pairs [14], whose mean value is $l = 7$ cm. When the new computed position of the agent would be outside of the tank, a new kick length is sampled from the distribution. The typical speed of fish right after a kick was found to be $v_0 \sim 14$ cm/s, and the speed was then found to decay exponentially during the gliding phase, with a relaxation time $\tau_0 = 0.8$ s (a feature implemented in the model in [14]). Thus, the duration of the time step $\tau_i^n$, updated at each kicking time of agent $i$, is determined by the length of the kick and the peak speed of the fish [14].

The variation of the heading angle of agent $i$ between two of its kicks is given by the sum of the variations induced by its environment, that is,

$$\delta\phi_i^n = \delta\phi_{w,i}^n + \delta\phi_{R,i}^n + \sum_{\langle j,i \rangle} \delta\phi_{ij}^n, \tag{4}$$

where $\delta\phi_{w,i}^n$ is the angular variation caused by static obstacles (the wall of the fish tank or the border of the robot platform), $\delta\phi_{R,i}^n$ is a random Gaussian white noise reflecting the spontaneous fluctuations in the motion of the agent, and $\delta\phi_{ij}^n$ is the angular variation induced by the social interaction of the agent $i$ with the agent $j$.

The notation $\langle j,i \rangle$ indicates that the sum is performed over all the agents $j$ considered to interact with agent $i$. The number $k$ of agents considered to interact with an agent is part of what constitutes a social interaction strategy, and remains constant along the whole simulation. When $k < N - 1$, the identity of these agents depends on the strategy, but also on the instantaneous state of the system, so that their identity must be updated at each kicking time of the focal agent $i$. At each kicking time $t_i^n$, the agents are sorted according to the criterion used in the interaction strategy: the distance to the focal agent $d_{ij}(t_i^n)$, a random selection of

neighbors, or the influence on the focal agent $\mathcal{I}_{ij}(t_i^n)$. Once sorted, the $k$ first agents are considered in the sum in Eq (4).

Each contribution to the angle variation can be expressed in terms of decoupled functions of the instantaneous state of the agents, that is, the distance and relative orientation to the wall $r_w$ and $\theta_w$, and the distance $d$, viewing angle $\psi$, and relative alignment $\phi$ between the focal fish and its considered neighbor (see Fig 2A). The derivation of these functions is based on physical principles of symmetry of the angular functions and a sophisticated reconstruction procedure detailed in Calovi *et al.* [14] for the case of *H. rhodostomus* and in [15] for the general case of animal groups.

For completeness, we show these functions in S9 Fig and present here their analytical expressions with the parameter values necessary to reproduce the simulations.

- The repulsive effect of the wall is a centripetal force that depends only on the distance to the wall $r_w$ and the relative angle of the heading to the wall $\theta_w$. Assuming that this dependence is decoupled, *i.e.*, $\delta\phi_w(r_w, \theta_w) = F_w(r_w)O_w(\theta_w)$, we have:

$$F_w(r_w) = \gamma_w \exp\left[-\left(\frac{r_w}{l_w}\right)^2\right], \quad O_w(\theta_w) = \beta_w \sin(\theta_w)(1 + 0.7\cos(2\theta_w)), \quad (5)$$

where $\gamma_w = 0.15$ is the intensity of the force ($F_w(0) = \gamma_w$), $l_w = 0.06$ m is the range of the wall repulsion, and $\beta_w = 1.9157$ is a normalization constant of the angular function $O_w(\theta_w)$, so that the mean of the squared function in $[-\pi, \pi]$ is equal to 1, that is, $(1/2\pi)\int_{-\pi}^{\pi} O_w^2(\theta)d\theta = 1$. All angular functions are normalized in this way, in order to allow the direct comparison of their shape in the different interactions.

These parameter values are those used in the model simulations. They also appear in Table 3, together with the values used in the experiments with robots.

- The intensity of the stochastic spontaneous variation of heading $\delta\phi_R$ depends on the distance to the wall $r_w$, and decreases as the fish gets closer to the wall and becomes constrained by the boundary of the tank:

$$\delta\phi_R(r_w) = \gamma_R\left(1 - \alpha\exp\left[-\left(\frac{r_w}{l_w}\right)^2\right]\right)g, \quad (6)$$

where $\gamma_R = 0.45$, $\alpha = 2/3$, and $g$ is a random number sampled from a standard normal distribution (zero mean; unit variance). Random variations are minimal at the border, where $r_w = 0$, $\delta\phi_R = \gamma_R(1 - \alpha)g$, and become larger as the individual moves away from the border, *i.e.*, as $r_w$ grows. Far from the border, the exponential goes to zero and $\delta\phi_R = \gamma_R g$.

- The interaction between agents can be decomposed into two terms of attraction and alignment which depend only on the relative state of both interacting agents:

$$\delta\phi_{ij}(d_{ij}, \psi_{ij}, \phi_{ij}) \quad = \quad \delta\phi_{Att}^{ij} + \delta\phi_{Ali}^{ij}, \quad (7)$$

$$= \quad \delta\phi_{Att}(d_{ij}, \psi_{ij}, \phi_{ij}) + \delta\phi_{Ali}(d_{ij}, \psi_{ij}, \phi_{ij}), \quad (8)$$

where the relative state of fish $j$ with respect to fish $i$ is given by $d_{ij}$, the distance between them; $\psi_{ij}$, the viewing angle with which fish $i$ perceives fish $j$; and $\phi_{ij} = \phi_j - \phi_i$, the difference between their heading angle.

We then define the *influence* $\mathcal{I}_{ij}(t)$ of a neighbor $j$ on a focal individual $i$ as the absolute contribution of this neighbor to the instantaneous heading change of the focal individual $\delta\phi_i(t)$

in Eq (4), that is, for $j = 1, \ldots, N, j \neq i$:

$$\mathcal{I}_{ij}(t) = |\delta\phi^{ij}_{\text{Att}}(t) + \delta\phi^{ij}_{\text{Ali}}(t)|. \tag{9}$$

This precise definition is central to the implementation of the MOST INFLUENTIAL interaction strategy involving the $k$ most influential neighbors of a given focal fish $i$ (*i.e.*, the $k$ neighbors with the largest influence $\mathcal{I}_{ij}(t)$).

Following [14], we assume that both the attraction and the alignment functions $\delta\phi^{ij}_{\text{Att}}$ and $\delta\phi^{ij}_{\text{Ali}}$ can be decomposed as the product of three functions that each depend on only one of the three variables determining the relative state of the two fish. Thus, for the attraction interaction, we have $\delta\phi_{\text{Att}}(d_{ij}, \psi_{ij}, \phi_{ij}) = F_{Att}(d_{ij})\, O_{\text{Att}}(\psi_{ij})\, E_{\text{Att}}(\phi_{ij})$, where

$$F_{\text{Att}}(d)\% = \gamma_{\text{Att}} \left( \frac{d/d_{\text{Att}} - 1}{1 + (d/l_{\text{Att}})^2} \right) = \gamma_{\text{Att}} \left( \frac{d}{d_{\text{Att}}} - 1 \right) \% \frac{1}{1 + \left(\frac{d}{l_{\text{Att}}}\right)^2} \frac{1}{1 + (d/l_{\text{Att}})^2}, \tag{10}$$

$$O_{\text{Att}}(\psi) = \beta_{\text{Att}} \sin(\psi)\left(1 - 0.33\cos(\psi)\right), \tag{11}$$

$$E_{\text{Att}}(\phi) = \lambda_{\text{Att}}\left(1 - 0.48\cos(\phi) - 0.31\cos(2\phi)\right). \tag{12}$$

Here, $d_{\text{Att}} = 3$ cm is the distance at which the short-range repulsion of individual collision avoidance balances the long-range repulsion, $\gamma_{\text{Att}} = 0.12$ is the intensity of the interaction, and $l_{\text{Att}} = 20$ cm characterizes the range where attraction is maximum. The angular functions $O_{\text{Att}}$ and $E_{\text{Att}}$ are respectively normalized with $\beta_{\text{Att}} = 1.395$ and $\lambda_{\text{Att}} = 0.9326$. As already mentioned when describing the interaction with the wall, the three functional forms defined in (10–12) and the numerical values of the coefficients have been extracted from experimental data by means of a sophisticated procedure based on physical principles of symmetry of the angular functions [14, 15]. The names of the angular functions stand precisely for their parity (Odd/Even).

In the alignment, we have $\delta\phi_{\text{Ali}}(d_{ij}, \psi_{ij}, \phi_{ij}) = F_{\text{Ali}}(d_{ij})\, E_{\text{Ali}}(\psi_{ij}) O_{\text{Ali}}(\phi_{ij})$, where

$$F_{\text{Ali}}(d) = \gamma_{\text{Ali}} \left( \frac{d}{d_{\text{Ali}}} + 1 \right) \exp\left[ -\left(\frac{d}{l_{\text{Ali}}}\right)^2 \right], \tag{13}$$

$$E_{\text{Ali}}(\psi) = \beta_{\text{Ali}}\left(1 + 0.6\cos(\psi) - 0.32\cos(2\psi)\right), \tag{14}$$

$$O_{\text{Ali}}(\phi) = \lambda_{\text{Ali}} \sin(\phi)\left(1 + 0.3\cos(2\phi)\right), \tag{15}$$

with $d_{\text{Ali}} = 6$ cm, $l_{\text{Ali}} = 20$ cm, $\gamma_{\text{Ali}} = 0.09$, $\beta_{\text{Ali}} = 0.9012$, $\lambda_{\text{Ali}} = 1.6385$.

The parameter values are those derived in [14] for the simulation model when fish swim in pairs and are summarized in Table 3 (fish model and robots). More details regarding the model, including the extraction of the above interaction functions, can be found in [14].

**Computational model in an unbounded domain.**   Model simulations of agents swimming in an unbounded domain were carried out by removing the interaction with the wall (*i.e.*, by setting $\gamma_{\text{w}} = 0$; the rest of parameter values being those given in Table 3).

We have considered the MOST INFLUENTIAL and NEAREST interaction strategies, that is, paying respectively attention to the $k$ most influential neighbors or to the $k$-nearest neighbors, for $k = 1, 2, 3,$ and 4, and the case where agents do not interact with each other ($k = 0$). Group cohesion and polarization are averaged over a large number of simulation runs $n$: $\langle C(t) \rangle = (1/n) \sum_{i=1}^{n} C_i(t)$, where $C_i(t)$ is the group cohesion at time $t$ in the $i$-th run. We used $n = 1000$. The duration of each simulation was sufficiently long to produce a total number of $10^4$ kicks per run among the 5 agents ($\sim 2.7$ hours). A second series of simulations was carried out to produce $5 \times 10^4$ kicks ($\sim 13.5$ hours), finding the same qualitative results. Initial conditions of each run were always different, with all agents located at less than $R = 25$ cm (the radius of the arena) from the origin of coordinates.

We first analyzed the impact on group cohesion and polarization (Figs 9 and 10) of reducing the attraction range in groups of $N = 5$ agents by truncating the attraction intensity function $F_{\text{Att}}$ when the neighbor is at a distance $d_{ij} > d_{\text{cut}}$ from the focal agent: $F_{\text{Att}}(d_{ij}) = 0$, if $d_{ij} > d_{\text{cut}}$. For each value of $d_{\text{cut}}$, the mean cohesion was calculated as the average over the last 10% of kicks over the 1000 runs carried out to obtain $\langle C(t) \rangle$, and this, for both considered strategies and each value of $k$. When $d_{\text{cut}}$ is sufficiently large, the attraction range is sufficiently long and $\langle C(t) \rangle$ is close to the value corresponding to the mean cohesion of the group when $F_{\text{Att}}$ is not truncated. When $d_{\text{cut}}$ is smaller than a critical cut-off $d_{\text{cut}}^*$, the attraction range is too short and the agents simply diffuse, with $\langle C(t) \rangle \sim t$ growing linearly in time Fig 9.

We then analyzed the group cohesion and polarization (Fig 10 and S7 Fig) $i$) in large groups of $N = 6, \ldots, 70$ agents for the MOST INFLUENTIAL strategy with $k = 1$, $ii$) in a group of size $N = 20$, for different values of the number of nearest neighbors $k$ with which agents interact, and $iii$) in groups of size $N = 5, \ldots, 26$, where agents interact with their $k$ nearest neighbors, for all the values of $k$ between 1 and $N - 1$, except for $N = 22, 24$ and 26, where we limited the simulations to the interval of interest $k = 8, \ldots, 12$. For each combination of group size $N$ and number of neighbors $k$ considered, the number of simulations, their duration, and the averaging procedure were the same as the ones used in the analysis of the groups of size $N = 5$.

## Quantification of the collective behavior

We characterize the collective behavioral patterns by means of five observables quantifying the behavior of the group in the tank and the behavior of individuals inside the group. We first write the coordinates of the position $\vec{u}_B = (x_B, y_B)$ and the velocity $\vec{v}_B = (v_x^B, v_y^B)$ of the barycenter $B$ (center of mass) of the group with respect to the reference system of the tank:

$$x_B(t) = \frac{1}{N} \sum_{i=1}^{N} x_i(t), \quad v_x^B(t) = \frac{1}{N} \sum_{i=1}^{N} v_x^i(t), \tag{16}$$

with similar expressions for $y_B(t)$ and $v_y^B(t)$. The heading angle of the barycenter is then given by $\phi_B = \text{atan2}(v_y^B, v_x^B)$.

The barycenter defines a system of reference in which the relative position and velocity of a fish, that we denote with a bar, are such that $\bar{x}_i = x_i - x_B$ and $\bar{v}_x^i = v_x^i - v_x^B$ (same expressions for the $y$-components). In the reference system of the barycenter, the angle of the position of a fish is given by $\bar{\theta}_i = \text{atan2}(\bar{y}_i, \bar{x}_i)$, so the relative heading in this reference system is $\bar{\phi}_i = \text{atan2}(\bar{v}_y^i, \bar{v}_x^i) \neq \phi_i - \phi_B$. We can thus define the angle of incidence of a fish with respect to a circle centered at the barycenter as $\bar{\theta}_w^i = \bar{\phi}_i - \bar{\theta}_i$. The angle $\bar{\theta}_w^i$ is the equivalent to the angle of incidence to the wall $\theta_w^i$ that we use in the reference system of the tank, and serves to measure the angular velocity of a fish with respect to the barycenter, in the reference system of the barycenter.

The five observables used to quantify the behavior of a group are defined as follows:

1. Group cohesion $C(t) \in [0, R]$:

$$C(t) = \sqrt{\frac{1}{N}\sum_{i=1}^{N} \| \vec{u}_i - \vec{u}_B \|^2}, \tag{17}$$

where $\| \vec{u}_i - \vec{u}_B \|$ is the distance from fish $i$ to the barycenter $B$ of the $N$ fish.
Low values of $C(t)$ correspond to highly cohesive groups, while high values of $C(t)$ (in particular, comparable to the radius of the tank) imply that individuals are spatially dispersed.

2. Group polarization $P(t) \in [0, 1]$:

$$P(t) = \frac{1}{N}\left\| \sum_{i=1}^{N}\vec{e}_i(t) \right\|, \tag{18}$$

where $\vec{e}_i = \vec{v}_i/\| \vec{v}_i \| = (\cos(\phi_i), \sin(\phi_i))$ is the unit vector in the direction of motion of the individual fish, given by its velocity vector $\vec{v}_i$.
A value of $P$ close to 1 would mean that the $N$ individual headings are aligned and point in the same direction, while a value of $P$ close to 0 would mean that the $N$ vectors point in different directions, but can also mean that vectors are collinear and with opposite direction (*e.g.*, for $N$ even, half of the vectors point North, the other half point South) so that they cancel each other. Similarly, when $N = 5$ and two normalized velocity vectors cancel each other (*e.g.*, when 4 fish swim in the same direction $\vec{e}$ and one fish swims in the opposite direction $-\vec{e}$) would give rise to a resultant vector of norm $P = (4 \times 1 - 1)/5 = 3/5 = 0.6$, and if two pairs of fish cancel each other, then $P = (3 \times 1 - 2 \times (-1))/5 = 1/5 = 0.2$.
Note that uncorrelated headings would lead to $P \sim 1/\sqrt{N}$, which becomes small only for large group size $N$, but which is markedly lower than 1 for any $N \geq 5$.

3. Distance of the barycenter to the wall $r_w^B(t) \in [0, R]$:

$$r_w^B(t) = R - \sqrt{\left(x_B(t)\right)^2 + \left(y_B(t)\right)^2}, \tag{19}$$

Note that when the individuals move in a cohesive group, $r_w^B$ is typically of the same order as the mean distance of agents to the wall $\langle r_w \rangle = (1/N)\sum_{i=1}^{N} r_w^i$. When the group is not cohesive, $r_w^B$ is of order of the radius of the tank.

4. Relative angle of the barycenter heading to the wall $\theta_w^B(t) \in [-\pi, \pi]$:

$$\theta_w^B(t) = \text{atan2}\left(v_y^B(t), v_x^B(t)\right). \tag{20}$$

When the group swims along the wall $\theta_w^B(t) \approx \pm\pi/2$ (*i.e.*, $\theta_w^B(t) \approx \pm 90°$).

5. Index of collective counter-milling and super-milling $Q(t) \in [-1, 1]$:

$$Q(t) = \left(\frac{1}{N}\sum_{i=1}^{N}\sin(\bar{\theta}_w^i((t)))\right) \times \text{sign}\left(\frac{1}{N}\sum_{i=1}^{N}\sin(\theta_w^i(t))\right) \tag{21}$$

$$= \Gamma_B(t) \times \text{sign}(\Gamma(t)). \tag{22}$$

A group of fish rotating around the center of the tank with a rotation index $\Gamma(t)$ (defined in Eq (22); similar to an angular momentum) would display a counter-milling behavior if the individual fish also rotate around the barycenter of the group and both directions of rotation are opposite. The first sum between parentheses in Eq (21) is the index of rotation of the fish with respect to the barycenter of the group, denoted by $\Gamma_B(t)$ in Eq (22). Multiplying by the sign of $\Gamma(t)$ means that when $Q(t) < 0$, both directions are opposite and the fish exhibit a *collective counter-milling behavior*, while when $Q(t) > 0$, both rotations are in the same direction and the fish exhibit a *collective super-milling behavior*.

Thus, a group of 5 individuals turning around the center of the tank in a rigid formation that always points North, like the fingertips of the hand when cleaning a window, would correspond to a perfect counter-milling behavior. On the other hand, a situation where individuals rotate around the center of the tank as if they were fixed to a vinyl record, so that trajectories are perfect circles and individuals far from the center of the tank move faster than those close to the center, would correspond to a zero-milling state. Actual groups of fish present an intermediate behavior between these two situations, with a clear bias towards negative values of $Q(t)$ (see Fig 3 for fish, S4 Video for robots, and Fig 8 for fish, model fish, and robots).

Collective behavior is thus quantified by means of the probability density functions of these quantities. In addition, density maps are presented in order to illustrate the correlations between the polarization $P$ and the group cohesion $C$ in fish experiments, model simulations, and robot experiments (S1–S4 Figs). We consider two normalizations: *i*) with the total number of data, to highlight the significant regions of the map and neglect the regions where the data are scarce (S1 Fig for the fish model, and S3 Fig for robot experiments); *ii*) with the total number of data in a given range of the polarization, so that each row in the map is a PDF of $C$ for a given $P$ (S2 Fig for the fish model, and S4 Fig for robot experiments). Spatial distances in the model and robot experiments are rescaled with the respective scaling factor $\lambda_M = 0.87$ and $\lambda_R = 0.35$ to allow for a direct comparison of our two spatial quantifiers ($C$ and $r_w^B$) with the results of fish experiments (the three other quantifiers $P$, $\theta_w^B$, and $Q$ are not affected by this rescaling).

## Quantifier for the similarity of probability distribution functions

In the Results section, we qualitatively compare the probability distribution functions (PDF) of the group cohesion, polarization, distance to the wall, angle with respect to the wall, and counter-milling index featured in Figs 4–8, for the 3 interaction strategies (NEAREST; RANDOM; MOST INFLUENTIAL), and for $k = 1, 2, 3$ interacting neighbors (as well as the cases $k = 0$—no interaction—and $k = 4$).

Here, we consider the Hellinger distance $D(F|G)$ [45, 46] to precisely quantify the "similarity" of two PDF $F(x)$ and $G(x)$ for the same observable $x$ (one of the 5 listed above that we have considered):

$$D(F|G) = \frac{1}{2} \int \left( \sqrt{F(x)} - \sqrt{G(x)} \right)^2 dx = 1 - \int \sqrt{F(x)} \sqrt{G(x)} \, dx, \tag{23}$$

where we have used the normalization of the PDF, $\int F(x) \, dx = \int G(x) \, dx = 1$, to obtain the last equality. The first definition of $D(F|G)$ makes clear that it measures the overall difference between $F(x)$ and $G(x)$, while the second equivalent definition has a nice interpretation in terms of the *overlap* of both PDF. Indeed, the second definition measures the distance from unity of the scalar product of $\sqrt{F(x)}$ and $\sqrt{G(x)}$ seen as vectors of unit Euclidean norm (a consequence of the normalization, $\int \sqrt{F(x)}^2 \, dx = 1$).

The Hellinger distance is zero if and only if $F(x) = G(x)$, and it always satisfies $D(F|G) \leq 1$. The upper bound $D(F|G) = 1$ is reached whenever the supports of the two PDF are not intersecting, so that $F(x) \times G(x) = 0$, for all values of $x$. In practice, a value of $D(F|G) \geq 0.1$ points to the two PDF being markedly dissimilar.

Of course, using the Hellinger distance is an arbitrary choice and other distances (like the Kolmogorov-Smirnov distance) could lead to slightly different relative distances/errors, but would not change our conclusions when the PDF are markedly different. In particular, the fact that the MOST INFLUENTIAL strategy is the strategy for $k = 1$ leading to the best agreement with fish experiments would be recovered by any meaningful quantifier.

We have computed the Hellinger distance between PDF measured in fish experiments and the corresponding PDF measured in the fish model simulations (Table 1) and in robots experiments (Table 2), hence providing a more precise, albeit not unique, quantification of their similarity.

## Supporting information

**S1 Fig. Density maps of the polarization vs cohesion for fish and model simulations, normalized with the total number of data.** Density maps are shown for fish experiments (FISH panel) and for the 11 strategies considered in the model simulations. The color intensity corresponds to the number of data in each box normalized with the total number of data in the grid ($\times 1000$). We used $40 \times 50$ boxes.
(TIF)

**S2 Fig. Density maps of polarization vs cohesion for fish and model simulations, normalized with the number of data per range of polarization.** Density maps are shown for fish experiments (FISH panel) and for the 11 strategies considered in the model simulations. The color intensity corresponds to the number of data in each box normalized with the number of data per interval of polarization, *i.e.*, each row is the PDF of the cohesion for a range of values of the polarization. We used $40 \times 50$ boxes.
(TIF)

**S3 Fig. Density maps of polarization vs cohesion for fish and robot groups, normalized with the total number of data.** Density maps are shown for fish experiments (FISH panel) and for the 10 strategies considered in the robot experiments. The color intensity corresponds to the number of data in each box normalized with the total number of data in the grid ($\times 1000$). We used $40 \times 50$ boxes.
(TIF)

**S4 Fig. Density maps of polarization vs cohesion for fish and robot groups, normalized with the number of data per range of polarization.** Density maps are shown for fish experiments (FISH panel) and for the 10 strategies considered in the robot experiments. The color intensity corresponds to the number of data in each box normalized with the number of data per interval of polarization, *i.e.*, each row is the PDF of the cohesion for a range of values of the polarization. We used $40 \times 50$ boxes.
(TIF)

**S5 Fig. Counter-milling in model simulations.** Red arrows represent the velocity field of agents in the reference system of the barycenter of the group, here located at coordinates $(0, 0)$. Orange circle denotes the average relative position of the border of the arena with respect to the barycenter. The cases where agents interact with the $k = 3$ most influential neighbors are statistically identical to the case where $k = 4$.
(TIF)

**S6 Fig. Counter-milling in robotic experiments.** Red arrows represent the velocity field of robots in the reference system of the barycenter of the group, here located at coordinates (0, 0). Orange circle denotes the average relative position of the border of the arena with respect to the barycenter. The cases where robots interact with the $k = 3$ most influential neighbors are statistically identical to the case where $k = 4$.
(TIF)

**S7 Fig. Average cohesion and polarization for group sizes $N = 5, \ldots, 20$ ($N$ even) when each individual interacts with its $k$ nearest neighbors, for $k = 1, \ldots N - 1$.** Mean cohesion (A) and mean polarization (B) as a function of $k$. Cohesion values are scaled with $\lambda_M = 0.87$. In panel (A), high values of the cohesion for small values of $k$ with respect to the group size $N$ grow up to 20 m in our simulations as the individuals diffuse independently of each other (vertical lines). In (B), the values of $k$ for $N = 22$, 24 and 26 (marked with an asterisk in the legend) are limited to the interval of interest [8, 12].
(TIF)

**S8 Fig. Finite state machine diagram of one robot.** The decision-making processes of the robot (COMPUTE state) are shown in blue. The movements of the robot (MOVE state) are shown in brown. In the COMPUTE state, the model determines a new target to reach by integrating the local information about the neighbors and the environment. A target is valid when this one is not blocked by the wall or other robots. If the target is invalid, the computer tries to find a new target by the scanning method. If the scanning fails, the robot moves back 80 mm and starts again for model computing. If the decision target is valid, the robot switches into MOVE state, which includes three sub-states: Rotate, Move straight, and Avoid obstacle. The robot first rotates towards to the target and then moves straight to it. If a running neighbor blocks the path, the robot uses a procedure to avoid the obstacle.
(TIF)

**S9 Fig. Interaction functions with the wall and between individuals, extracted from experiments of fish swimming in pairs [14].** (A) Intensity of the repulsion from the wall $F_w(r_{w,i})$ (green) as a function of the distance to the wall $r_w,i$, and intensity of the attraction $F_{Att}(d_{ij})$ (red) and the alignment $F_{Ali}(d_{ij})$ (blue) between fish $i$ and $j$ as functions of the distance $d_{ij}$ separating them. (B) Normalized odd angular function $O_w(\theta_{w,i})$ modulating the interaction with the wall as a function of the relative angle to the wall $\theta_{w,i}$. (C) Normalized angular functions $O_{Att}(\psi_{ij})$ (odd, in red) and $E_{Att}(\phi_{ij})$ (even, in orange) of the attraction interaction, and (D) $O_{Ali}(\phi_{ij})$ (odd, in blue) and $E_{Ali}(\psi_{ij})$ (even, in violet) of the alignment interaction between agents $i$ and $j$, as functions of the angle of perception $\psi_{ij}$ and the relative heading $\phi_{ij}$.
(TIF)

**S1 Video. Collective movements in rummy-nose tetra (*Hemigrammus rhodostomus*).** A typical experiment with a group of 5 fish swimming in a circular tank of radius 250 mm.
(MP4)

**S2 Video. Collective motion in a group of 5 robots.** Each robot interacts with its most influential neighbor. The video is accelerated 9 times. Total duration: 7.15 minutes.
(MP4)

**S3 Video. Tracking and analysis output.** The small circles superimposed on the trajectories represents the kicks performed by the fish when the speed reaches its maximum value.
(MP4)

**S4 Video. Counter milling behavior in a group of 5 fish.** Top: Typical experiment with a group of 5 fish in a circular arena of radius 250 mm. The video is accelerated 6 times. Total duration 1.3 minutes. Bottom: Relative movement of fish with respect to the barycenter of the group, represented by the black arrow on top video and a black disk on the bottom video. Fish turn counter-clockwise around the tank and clockwise with respect to the barycenter. (MP4)

**S5 Video. Collective robotics experiment without any social interaction between the robots ($k$ = 0) and only obstacle avoidance behavior is at play.** Top: Typical experiment with a group of 5 robots in a circular arena of radius 420 mm, captured by the top camera. The border of the arena is represented by the red circle. Purple circles represent the individual robot safety area, of diameter 8 cm. Small green dots in front of robots indicate their next target place. The video is accelerated 6 times. Total duration: 6 minutes. Bottom: Relative movement of the robots with respect to the barycenter of the group. The barycenter is represented by the black disk and remains oriented to the right. Robots are represented by colored disks with their identification number in the center. The small circle at the front of a robot indicates its heading. The arrows represent the interactions between robots. Arrow direction indicates the identity (color) of the robot that exerts its influence on the robot to which the arrow points. The small dots in front of the robots represent the next target places. (MP4)

**S6 Video. Collective robotics experiment where robots interact with the $k$ = 1 nearest neighbor.** Top: Typical experiment with a group of 5 robots in a circular arena of radius 420 mm, captured by the top camera. The border of the arena is represented by the red circle. Purple circles represent the individual robot safety area, of diameter 8 cm. Small green dots in front of robots indicate their next target place. The video is accelerated 6 times. Total duration: 6 minutes. Bottom: Relative movement of the robots with respect to the barycenter of the group. The barycenter is represented by the black disk and remains oriented to the right. Robots are represented by colored disks with their identification number in the center. The small circle at the front of a robot indicates its heading. The arrows represent the interactions between robots. Arrow direction indicates the identity (color) of the robot that exerts its influence on the robot to which the arrow points. The small dots in front of the robots represent the next target places. (MP4)

**S7 Video. Collective robotics experiment where robots interact with the $k$ = 1 most influential neighbor.** Top: Typical experiment with a group of 5 robots in a circular arena of radius 420 mm, captured by the top camera. The border of the arena is represented by the red circle. Purple circles represent the individual robot safety area, of diameter 8 cm. Small green dots in front of robots indicate their next target place. The video is accelerated 6 times. Total duration: 6 minutes. Bottom: Relative movement of the robots with respect to the barycenter of the group. The barycenter is represented by the black disk and remains oriented to the right. Robots are represented by colored disks with their identification number in the center. The small circle at the front of a robot indicates its heading. The arrows represent the interactions between robots. Arrow direction indicates the identity (color) of the robot that exerts its influence on the robot to which the arrow points. The small dots in front of the robots represent the next target places. (MP4)

**S8 Video. Collective robotics experiment where robots interact with $k$ = 1 randomly selected neighbor.** Top: Typical experiment with a group of 5 robots in a circular arena of

radius 420 mm, captured by the top camera. The border of the arena is represented by the red circle. Purple circles represent the individual robot safety area, of diameter 8 cm. Small green dots in front of robots indicate their next target place. The video is accelerated 6 times. Total duration: 6 minutes. Bottom: Relative movement of the robots with respect to the barycenter of the group. The barycenter is represented by the black disk and remains oriented to the right. Robots are represented by colored disks with their identification number in the center. The small circle at the front of a robot indicates its heading. The arrows represent the interactions between robots. Arrow direction indicates the identity (color) of the robot that exerts its influence on the robot to which the arrow points. The small dots in front of the robots represent the next target places.
(MP4)

**S9 Video. Collective robotics experiment where robots interact with the $k = 2$ nearest neighbors.** Top: Typical experiment with a group of 5 robots in a circular arena of radius 420 mm, captured by the top camera. The border of the arena is represented by the red circle. Purple circles represent the individual robot safety area, of diameter 8 cm. Small green dots in front of robots indicate their next target place. The video is accelerated 6 times. Total duration: 6 minutes. Bottom: Relative movement of the robots with respect to the barycenter of the group. The barycenter is represented by the black disk and remains oriented to the right. Robots are represented by colored disks with their identification number in the center. The small circle at the front of a robot indicates its heading. The arrows represent the interactions between robots. Arrow direction indicates the identity (color) of the robot that exerts its influence on the robot to which the arrow points. The small dots in front of the robots represent the next target places.
(MP4)

**S10 Video. Collective robotics experiment where robots interact with the $k = 2$ most influential neighbors.** Top: Typical experiment with a group of 5 robots in a circular arena of radius 420 mm, captured by the top camera. The border of the arena is represented by the red circle. Purple circles represent the individual robot safety area, of diameter 8 cm. Small green dots in front of robots indicate their next target place. The video is accelerated 6 times. Total duration: 6 minutes. Bottom: Relative movement of the robots with respect to the barycenter of the group. The barycenter is represented by the black disk and remains oriented to the right. Robots are represented by colored disks with their identification number in the center. The small circle at the front of a robot indicates its heading. The arrows represent the interactions between robots. Arrow direction indicates the identity (color) of the robot that exerts its influence on the robot to which the arrow points. The small dots in front of the robots represent the next target places.
(MP4)

**S11 Video. Collective robotics experiment where robots interact with $k = 2$ randomly selected neighbors.** Top: Typical experiment with a group of 5 robots in a circular arena of radius 420 mm, captured by the top camera. The border of the arena is represented by the red circle. Purple circles represent the individual robot safety area, of diameter 8 cm. Small green dots in front of robots indicate their next target place. The video is accelerated 6 times. Total duration: 6 minutes. Bottom: Relative movement of the robots with respect to the barycenter of the group. The barycenter is represented by the black disk and remains oriented to the right. Robots are represented by colored disks with their identification number in the center. The small circle at the front of a robot indicates its heading. The arrows represent the interactions between robots. Arrow direction indicates the identity (color) of the robot that exerts its influence on the robot to which the arrow points. The small dots in front of the robots

represent the next target places.
(MP4)

**S12 Video. Collective robotics experiment where robots interact with the $k = 3$ nearest neighbors.** Top: Typical experiment with a group of 5 robots in a circular arena of radius 420 mm, captured by the top camera. The border of the arena is represented by the red circle. Purple circles represent the individual robot safety area, of diameter 8 cm. Small green dots in front of robots indicate their next target place. The video is accelerated 6 times. Total duration: 6 minutes. Bottom: Relative movement of the robots with respect to the barycenter of the group. The barycenter is represented by the black disk and remains oriented to the right. Robots are represented by colored disks with their identification number in the center. The small circle at the front of a robot indicates its heading. The arrows represent the interactions between robots. Arrow direction indicates the identity (color) of the robot that exerts its influence on the robot to which the arrow points. The small dots in front of the robots represent the next target places.
(MP4)

**S13 Video. Collective robotics experiment where robots interact with $k = 3$ randomly selected neighbors.** Top: Typical experiment with a group of 5 robots in a circular arena of radius 420 mm, captured by the top camera. The border of the arena is represented by the red circle. Purple circles represent the individual robot safety area, of diameter 8 cm. Small green dots in front of robots indicate their next target place. The video is accelerated 6 times. Total duration: 6 minutes. Bottom: Relative movement of the robots with respect to the barycenter of the group. The barycenter is represented by the black disk and remains oriented to the right. Robots are represented by colored disks with their identification number in the center. The small circle at the front of a robot indicates its heading. The arrows represent the interactions between robots. Arrow direction indicates the identity (color) of the robot that exerts its influence on the robot to which the arrow points. The small dots in front of the robots represent the next target places.
(MP4)

**S14 Video. Collective robotics experiment where robots interact with all their neighbors ($k = 4$).** Top: Typical experiment with a group of 5 robots in a circular arena of radius 420 mm, captured by the top camera. The border of the arena is represented by the red circle. Purple circles represent the individual robot safety area, of diameter 8 cm. Small green dots in front of robots indicate their next target place. The video is accelerated 6 times. Total duration: 6 minutes. Bottom: Relative movement of the robots with respect to the barycenter of the group. The barycenter is represented by the black disk and remains oriented to the right. Robots are represented by colored disks with their identification number in the center. The small circle at the front of a robot indicates its heading. The arrows represent the interactions between robots. Arrow direction indicates the identity (color) of the robot that exerts its influence on the robot to which the arrow points. The small dots in front of the robots represent the next target places.
(MP4)

## Acknowledgments

We are grateful to Patrick Arrufat and Gérard Latil for technical assistance.

## Author Contributions

**Conceptualization:** Liu Lei, Clément Sire, Guy Theraulaz.

**Data curation:** Liu Lei, Ramón Escobedo.

**Formal analysis:** Liu Lei, Ramón Escobedo.

**Funding acquisition:** Liu Lei.

**Investigation:** Liu Lei, Ramón Escobedo, Clément Sire, Guy Theraulaz.

**Methodology:** Liu Lei, Ramón Escobedo.

**Project administration:** Guy Theraulaz.

**Software:** Liu Lei, Ramón Escobedo.

**Supervision:** Clément Sire, Guy Theraulaz.

**Validation:** Liu Lei, Ramón Escobedo, Guy Theraulaz.

**Visualization:** Liu Lei, Ramón Escobedo.

**Writing – original draft:** Liu Lei, Ramón Escobedo, Guy Theraulaz.

**Writing – review & editing:** Liu Lei, Ramón Escobedo, Clément Sire, Guy Theraulaz.

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
