## [Decision Letter · Decision Letter 0]

29 Jul 2019

Dear Dr Theraulaz,

Thank you very much for submitting your manuscript 'Computational and robotic modeling reveal parsimonious combinations of interactions between individuals in schooling fish' for review by PLOS Computational Biology. Your manuscript has been fully evaluated by the PLOS Computational Biology editorial team and in this case also by independent peer reviewers. The reviewers appreciated the attention to an important problem, but raised some substantial concerns about the manuscript as it currently stands. While your manuscript cannot be accepted in its present form, we are willing to consider a revised version in which the issues raised by the reviewers have been adequately addressed. We cannot, of course, promise publication at that time.

Sincerely,

Vito Trianni, Ph.D.

Guest Editor

PLOS Computational Biology

Samuel Gershman

Deputy Editor

PLOS Computational Biology

[LINK]

The paper requires major revisions in order to address all the comments made by the reviewers, which are valid and should be carefully considered. Special attention should be paid to the following:

1. Although the "most influential neighbour" hypothesis can explain the observations from experiments, it is not fully justified from a cognitive point of view (one reviewer talks about 'self-referential' definition, another one about the computational load required to individuals to identify the most-influential neighbour), and neither the computational nor the robotic model seems to address this point.

2. There is a lack of quantitative comparison between the results obtained with different models. This is particularly important to support the authors' claims.

3. The contribution obtained from implementing the model in a (seemingly centralised) robotic system is not clear, and should be better explained

Reviewer's Responses to Questions

**Comments to the Authors:**

Reviewer #1: In the paper “Computational and robotic modeling reveal parsimonious combinations of interactions between individuals in schooling fish” the authors study collective motion in fish by means of mathematical modeling and using swarm robots as models. The main focus is on how many neighbors need to be taken into consideration for an individual in order to generate a valid collective motion with appropriate cohesion and polarization.

The subject is relevant and within scope. The paper is overall well written and relatively easy to follow. Several logical entities are not explained and discussed in one place but are rather distributed over the whole paper. As a consequence a few aspects are difficult to understand, especially for a reader new to collective motion in fish (details below). Another problem is that the overall contribution of the paper is a bit difficult to assess although the finding of small neighbor selections being effective is relevant. A third issue is that neither the robot experiments themselves nor the methodology of the robot experiments are sufficiently described.

A - distributed and incomplete explanations

The description of the robot methodology is distributed over at least two sections, imprecise, and incomplete.

A crucial term is that of “kick” which is frequently used before it is vaguely defined. Please introduce the term before using it and try to define it as accurate as possible. Since you are using an automated kick detection to process the biological data, it may make sense to report that in connection to that definition. Similarly, “kick length” should be defined more clearly - possibly as displacement or covered distance.

The description of most figures is distributed over the whole results section. If a reader wants to look up the text for a given figure, then that is basically impossible.

B - contribution

It is a bit difficult to assess the significance of the contribution. There is a contribution and it is relevant but more discussion is required. How does this work relate to your own previous publications and what are the most closely related papers of other groups? The introduction is great to read and gives a good overview of related work but those papers are then not directly compared to this paper’s contribution.

C - robot experiments

It seems likely to me that the robots are completely controlled centrally and externally. It seems that the robot uses none of its own sensors. However, that remains crucially unclear because it is mentioned that the robots are “perfectly autonomous” and (only?) the decision-making is externalized. The use or non-use of sensors is not discussed. A better description of Fig. 12 within the text may help here. It remains also unclear whether or how frequent kicks of the robots are enforced given that robots need to turn on spot often. How do you, for example, prevent that a robot just keeps turning for long as the requirements for to where turn change while it is turning?

There is basically no text about how the robot experiments were done. The platform and setup are described but not how many experiments under what conditions were executed for how long etc.

The necessity of the robot experiments is not well motivated, even more so if the robot control is centralized and robots are only remotely controlled. What are specific embodied features that you can test in this robot setup?

It would have been easier to review, if the figures would have been embedded in the text.

For a possible revision, please mark all your changes in color.

Detailed comments

1 - abstract, “data-based”: To me it is unclear in what sense the model can be described as data-based.

2 - abstract, “only with the neighbor having the strongest effect on its heading variation”: This can be misread as a self-referential definition. In fact, there is no better definition in the whole paper. It is only repeated twice later into the paper.

3 - p. 3, “brings us back to a often”: to _an_ often

4 - results, six quantities: I understand that the main definition follows in the method section; still, a natural language definition in half a sentence per quantity would be useful.

5 - p. 4&5, counter-milling: From the explanation in the text one could interpret counter-milling as an artifact of the experiment setup. It what sense is it a relevant and/or natural behavior?

6 - p. 5, “formation of group”: formation of _a_ group

7 - p. 7, Fig. S6: Figure and data are difficult to understand based on the little provided information

8 - p. 7, “scaling factor” lambda: The concept of the scaling factor is difficult to understand based on the given information.

9 - p. 10, description of “most influential neighbor”: concept still unclear and seemingly self-refential

10 - p. 11, “150 L”: 150 l?

11 - p. 11, “box to isolate”: unclear, what kind of box?

12 - p. 12, “Cuboids … they are 60 m high”: seems not right

13 - p. 12, “messages infrared signals”: please correct

14 - p. 12, “neighboring robot that transmit”: please correct

15 - p. 12, “by triangulation method”: please correct

16 - p. 13, “kind of bubble sort algorithm”: This is maybe not the best comparison; also bubble sort is considered to be one of the worst sorting algorithms

17 - p. 14, “…is the kick length”: Please give a comprehensive definition.

18 - p. 16, “being those given in Table 1”: missing parenthesis

19 - p. 17, “among the robot”: among the _robots_

Reviewer #2: This article contributes to the important line of research aiming at revealing the nature of cognitive and interaction mechanisms used by social animals to perform collective motion. In this work, the authors analyze collective motion data of a group of 5 H. rhodostomus fish (by computing a set of group-level metrics), and develop two synthetic models to test their understanding on the modalities of group interactions: a computational model and a swarm robotics model. The main finding of the paper is that individual fish need only to interact with one "influential" neighbor to achieve good level of coherence and order, and that interacting with two neighbors (no matter how chosen) is enough for the models to replicate closely the data of the fish.

This paper is an important addition to the literature and I would like to see it published. I have however a number of moderate to high level concerns that apply to the manuscript in its current form:

- In the introduction, paragraph starting at line 98, the author claim that their goal is to "better understand how individual combine and integrate interactions with their neighbors in a group of moving animals". However, the study is mainly focused on the number of neighbors the focal individual needs to interact with, not really on the mechanism by which information combing from neighbors is combined. A similar statement that needs to be slightly aligned with the results is the one at lines 109-113 (it is not clear here whether one needs one neighbors or two or three, while later it is clear that one neighbor is enough to reproduce collective motion and two-three to reproduce the data). I believe these statements in the introduction should be revised to achieve alignment between contribution statement and results.

- I believe some essential details to assess the importance of this contribution are located in the materials and methods section, however they should briefly outlined in the main body of the paper as well. In the study of collective motion, two important questions are what type of information is exchanged between individuals (e.g. orientation only vs relative position and bearing only vs both), and how this information is integrated (see my previous point). The computational model does contain detail about those, from my understanding both angular as well as positional/bearing information are assumed to be exchanged, and a sort of averaging process is assumed. I do agree that the full details of the model should not be revealed in the main body, however the authors should at least mention these two pieces of information (what information is exchanged and how it is integrated). Also some high level details on how the most influential neighbors are determined should be indicated in the main body. To be honest, the full details of this are not entirely clear to me even after reading the materials and method, perhaps because this is the detailed subject of another publication. The type of information that should be present in the main body is: are influential neighbors determined at every timestep (therefore the topology is dynamic), or only at the beginning of each experiment (fixed topology)? are those determined based on the computational model ? (from my understanding the computational model is "simulated" by faking the interaction with all the other agents, determining the effect of each interaction and only choosing the k neighbors that have the largest effect, if my understand is right a description of this sort, very high level, should I believe be included in the main body).

- Linked to the previous point, I think the related work is missing few references that are strongly related. For example, these two papers and possibly others (dealing with the interaction topology in fish) could be referenced in the discussion at paragraph line 59 to 84:

https://www.pnas.org/content/112/15/4690

https://www.ncbi.nlm.nih.gov/pubmed/21795604

while other models that do include physical constraints (e.g. those only based on attraction repulsion that consider non-instantaneous relaxation of the heading direction) could be mentioned as those have also been tested on robotics platforms:

https://link.springer.com/article/10.1007/s10955-014-1114-8

https://journals.sagepub.com/doi/10.1177/1059712312462248

- When assessing whether both models achieve good matching with the experimental data, it would be great if the authors could devise a method to quantify this numerically, for example a measure of the overlap between the two distribution (empirical vs model) for each of the metric. If this can be done it would make the description of the results much stronger and it would be possible also to support claims on what type of interaction better reproduces the data, on when there is no longer a statistically significant difference (e.g. between k=2 and k=3, etc ...).

- I find the section collective motion in unbounded space quite important to assess the generality of the findings. Here, the paper could be made much stronger by testing the models on different conditions, most importantly on much larger school sizes, to test scalability. Although it is understandable that the data is restricted for practical reason to N=5, the computational model can be instead tested on much larger swarms. For example, I would be very curious to know if the mechanism that relies only on the most influential neighbor is sufficient to generate collective motion for large schools too.

- Still in the section on unbounded space, I think it is important here to report and discuss the degree of polarization, as in this setting this may be the most important metric.

- I found the Discussion section quite weak, in the sense that it mainly re-iterates through all the results, in less details, but roughly in the same way as the Results section. I think the Discussion section should instead focus on discussing the wider implications of these results, and relating this to what is already known in the field. Having said this, I do admit that the description of the results (as done in the discussion) was indeed very useful, so this should not be discarded, but the authors should avoid discussing the results twice (between the two I would keep the result description currently done in Discussion and not the one done in Results).

- The school size N=5 is declared quite late in the paper, but is needed from line 127 onwards, where I kept on wondering why only k=1, 2, 3 etc were used and kept on asking myself how large was the total school.

- Line 225-226: this sentence is quite vague, it would be best to specify which metrics are being referred too here (all of them?)

- Line 557 onwards: please discuss more the details of the triangulation method, because I could not understand how this was done with only two measurements (normally at least 3 are required).

- Line 666: is the ref to Fig1C here right ? (it is showing the fish trajectory and not the "decision process of the agent".

- What is the meaning of the three components of decoupling of the computational model? (Eqn 8 to 13). The authors should briefly outline what each of these three functions are for the two components of the model.

- Line 756 (and also in other places afterwards): how does being Object-Oriented Programming qualify the software platform developed for Cuboids ? I do not object the fact that OOP was used instead of simple procedural programming, however I do not understand how this qualifies the type of architecture, and why this was chosen as a name.

- Putting aside the extra complications arising from having a physical platform (obstacle avoidance, finite state machine for achieving the kicking behavior etc), is the model used with Cuboids an adaptation of the same computational model studied earlier? This should be specified (even in the main text), even if very briefly.

- Information contained in the section starting at 823 comes somehow too late and contains some overlapping information with what was said before, with some important details added. I think this section should disappear and the content be merged and those important details added earlier where relevant. Or the section could be moved earlier and include also the information presented somewhere else (for example how to calculate the most influential neighbors).

Reviewer #3: The manuscript “Computational and robotic modeling reveal parsimonious combinations of interactions between individuals in schooling fish” implements rules of interaction previously deciphered between pairs of fish into a model simulation and robotic swarm with the aim of determining the structure of the interactions in real fish. The paper seems to be the first attempt at taking rules directly inferred from real animals and implementing them into robot systems. This makes the paper particularly novel and interesting and will make an important contribution to self-organization, bio-inspiration and collective behavior. However, there are a number of important issues that I would like to raise before the paper can be accepted for publication.

General comments:

While I was very impressed by the interdisciplinary approach to the topic, I was a little disappointed with the lack of statistical comparison between the real data and the simulation and robotic results. Many statements are made without any comparative approaches, and in some cases, my interpretation of the results appears different from the authors. It would relatively simple to implement tests such as Kolmogorov-Smirnov tests, that can statistically compare the distributions of the variables (PDFs) that the authors have computed between the fish, model, and robots. This would allow the authors to make more quantitative comparisons and to more accurately identify which strategies capture the real fish’s behaviour.

It would be particularly useful to note the best-supported model (i.e. nearest, random, influential, for k = 1, 2,3 etc.) for each measure that the authors calculate. For example, the authors could identify in a table the (statistically) best-supported model for group cohesion, group polarization, mean distance of individuals to the border, etc. This would allow readers to identify consistently (across different metrics) chosen models that are supported by the data. As the authors suggest, this could be models that only use one or two influential neighbours, but this needs to be tested. As the manuscript currently stands, readers cannot integrate all the information that the authors have produced in order to determine which models are superior over others.

I also found the paper particularly difficult to follow in places when figures, and importantly the subpanels within the figures, were not referred to in the results section. For example, lines 208-211 discuss polarization without a reference to the polarization figure. This is not the only example. There were also many relevant and important lines (curves) missing from the figures, which made interpretation impossible (see my specific comments below).

The authors argue in the discussion that responding to a small number of neighbours could represent selection attention mechanisms, thereby reducing the amount of information that an animal needs to process. However, from my understanding, the implementation of the ‘most influential’ neighbour rule appears to require a focal individual to calculate the influence that all neighbours have on that individual, and then choose the neighbour with the most influence. This does not, therefore, reduce the information an individual needs to process. I would like to see the authors address this point in the manuscript.

In relation to the most influential neighbour, it appears from Movie S7 that for large portions of the time (perhaps the majority of the time) most individuals are following a single neighbour - the neighbour that is closest to the wall and perhaps making the largest contributions towards angular differences between the robots. Is this also observed in the model simulations? Could the authors discuss what implications this could have on distributed decision-making, if particular individuals are disproportionally influencing collective movement?

In general, I would also like the authors to be careful with some terminology. The authors refer to groups being ‘better’ or ‘worse’ in terms of cohesion or polarization. Please simply state the direction of effects.

Other comments:

Line 109: Remove “a”.

Line 112: Change “basically” with (i.e. one or two).

Line 147: remove “huge”

Line 228:230: This statement does not seem to be supported by the data. Counter-milling is observed when individuals interact with a nearest neighbour only. Figure 9 shows evidences of this, as does Fig. S5. Please rephrase.

Line 235: The line of k = 1 for the most influential neighbour in Figure 6C appears to be missing, making this comparison impossible.

Line 236: I do not see any evidence that if a fish interacts with a random nearest neighbour, that group cohesion is ‘better’ than if they interacted with a nearest neighbour. There does not appear to be any differences in the shape of the curves in Fig. 4 A and B. Further, Fig. S5 appears to be a referring to the wrong figure here. Again, statistical comparison would aid in data interpretation here.

Line 248: Again, this statement does not appear to be supported by the data. Groups are more cohesive and more polarised when k = 2, than when k = 1 across all strategies.

Line 250: Coincide more frequently with what?

Line Again, the line appears to be missing for k = 3 appears to be missing from Fig. 9c.

Line 364: There doesn’t appear to be any groups that are larger than 18 cm in Fig. 4D? Similarly there are no groups that are further than 22 cm from the border in Fig. 6D.

Figure 10A: Again, the line for k = 0 appears to be missing from the plot. This may be because cohesion reduces rapidly, but then this should referred to in the caption. The scale on the x axis for subplots D & E should be the same.

Line 530: Were the experimental fish used more than once? Please add details.

Line 527: 60 mm not 6 m !

Line 558: ‘Peering angle’ is undefined, and doesn’t appear to be depicted in Fig. 2. Please clarify.

Line 564-565: This sentence is unclear. How many time-steps is required before a neighbour’s id is deleted from the list?

Robots and Experimental Platform sections: It is not clear to me whether the robots are using the infra-red signals from neighbouring robots to inform their decisions to move. Given that a whole section (lines 553 – 572) is dedicated to explaining robot-robot sensing, I would suspect so, but it is not clear given that the computer software is providing the robots with control signals. These sections, along with Figures 12 and 13 need significant clarification. To start, it may help moving paragraph beginning line 595 to the start of the experimental platform section. It may also help explicitly stating that infra-red communication is not-used (if it isn’t) to inform individual robots movements when describing this earlier on.

Figures 12 and 13 would benefit from consistent terminology and highlighting what the coloured regions represent. i.e. What does the orange section of this figure represent? What is the red section?

Figure 13: “for each robot based [on] its local information”

Line 629: mins not (mn).

Line 631: Remove this sentence as the kick are not independent and therefore the this is not 16 hrs of independent data, which it could be misconstrued as.

Line 634: Surely the local minima (not maximum) is defined as the onset of a kick?

Line 637: The authors ‘double their data’ by creating the ‘mirror-image’ of the trajectories. This represents a form of pseudo-replication and should be avoided. Why can't the authors simply compare the absolute distributions (e.g. absolute rotation index between the real data and the simulations/robotic results)?

Line 660: “Decisions of the fish in the simulations and the decisions of the robots”

Figure 2: line r_wi is not horizontal. Psi_ij is not represented in the figure (Psi_ji is, not this is not referred to in the caption).

**Have all data underlying the figures and results presented in the manuscript been provided?**

Reviewer #1: Yes

Reviewer #2: No: It says data will be available after publication in a public repository, no further details or URL are provided

Reviewer #3: No: All data should be provided via a public repository.

PLOS authors have the option to publish the peer review history of their article (what does this mean?). If published, this will include your full peer review and any attached files.

Reviewer #1: No

Reviewer #2: Yes: Eliseo Ferrante

Reviewer #3: No

---

## [Decision Letter · Decision Letter 1]

18 Dec 2019

Dear Dr Theraulaz,

Thank you very much for submitting your manuscript, 'Computational and robotic modeling reveal parsimonious combinations of interactions between individuals in schooling fish', to PLOS Computational Biology. As with all papers submitted to the journal, yours was fully evaluated by the PLOS Computational Biology editorial team, and in this case, by independent peer reviewers. The reviewers appreciated the attention to an important topic but identified some aspects of the manuscript that should be improved.

We would therefore like to ask you to modify the manuscript according to the review recommendations before we can consider your manuscript for acceptance. Your revisions should address the specific points made by each reviewer and we encourage you to respond to particular issues Please note while forming your response, if your article is accepted, you may have the opportunity to make the peer review history publicly available. The record will include editor decision letters (with reviews) and your responses to reviewer comments. If eligible, we will contact you to opt in or out.raised.

- Supporting Information uploaded as separate files, titled 'Dataset', 'Figure', 'Table', 'Text', 'Protocol', 'Audio', or 'Video'.

We hope to receive your revised manuscript within the next 30 days. If you anticipate any delay in its return, we ask that you let us know the expected resubmission date by email at ploscompbiol@plos.org.

Sincerely,

Vito Trianni, Ph.D.

Guest Editor

PLOS Computational Biology

Samuel Gershman

Deputy Editor

PLOS Computational Biology

[LINK]

The reviews agree on the substantial improvements on the text, but few amendments are still necessary. In particular, pay attention to define properly the contribution of the paper, and to the motivations and contributions provided by the usage of a centralised multi-robot system.

Reviewer's Responses to Questions

**Comments to the Authors:**

Reviewer #1: The authors have carefully prepared an extensive revision of their paper. The quality has certainly increased and many of my previous concerns have been resolved. I still see a few issues that would need to be resolved.

1 - “most influential strategy” with k=1 neighbors: Thanks to the improved paper I have a better understanding of the “most influential” strategy now. What I had flagged as “potentially self-referential” before is not yet resolved but I can clarify my concerns now. What’s still unclear from the text, I think, are the temporal aspects of the strategy and what information needs to be available at what time. The mathematical description (eqs. 4 and 9) could be improved accordingly.

1a - most influential over what time period?

It would be possible that a fish picks a neighbor that is most influential over its lifetime, experiment duration, or a single time step (whatever that would biologically mean), etc. Most influential during lifetime and experiment could only be determined posteriori, hence is probably not useful. I believe you mean most influential in the current time step. This is only implicitly stated in the paper (or I missed it). Please clarify. I’d recommend to extend the mathematical description, such that an explicit maximization operation over all neighbors for a current time step is given. Something like: for given time step t most influential neighbor j of fish i: j(t) = arg max_j I_{ij}(t)

1b - most influential neighbor selection is valid for how long?

It would be possible that a fish picks a most influential neighbor for a duration of its lifetime, experiment duration, or a single/multiple time steps. I believe it’s only valid for 1 time step. However, that means that each fish i has to check each potential neighbor j in each time step t whether it is its potentially most influential neighbor; which leads to issue 1c.

1c - is the concept of k=1 misleading?

Often in the paper you seem to argue that the most influential neighbor strategy is efficient/filtering (?) in a sense that only one neighbor is considered (e.g., line 109 “interacts only with a single neighbor"). However, each fish needs to consider all other fish in each time step to find its currently most influential neighbor. In terms of computational complexity, what does not scale is the mere consideration of all neighbors/swarm members independent of the actual calculation as given by eq. 4. Please clarify. Please also discuss that this strategy would not scale. Possibly, it makes sense to think of a two-step selection process: first selection by distance and then selection by most influential? Scaling is certainly not relevant for the investigated swarm size of N=5 but would be in general.

2 - central control and remote controlled robots:

Now it is clearer that the robots are non-autonomous and remote controlled. Could you please state that very clearly in the paper? You correctly mention that Fig. 13 reflects that but Fig. 12 may even be better for that. Maybe you can indicate there more clearly that the communication by Wifi is one-way (if it actually really is) from the computer to the robots. Hence, no sensors of the robots used and the robots compute nothing onboard.

3 - embodiment:

You have added arguments of why an embodied model in the form of swarm robots is helpful. I’m certainly on your side here but the mentioned issues of collision avoidance and time-consuming turns on the spot due to non-holonomic drives could actually be simulated. Maybe it is also about imperfections in the robots and influences of the real world? Noisy sensing seems mostly excluded due to the central sensing approach.

4 - real time:

You are referring to “real time” several times. It is a bit unclear whether this is meant as “real-time control” such that you can guarantee that (each robot?) reacts within strict max. time intervals.

Reviewer #2: The authors have now produced a much improved version of the manuscript, addressing in a careful and satisfactory way all the comments of the three referees. From my side the article is very close to publication quality. Only few comments left:

- My very first remark about how the contribution is claimed was partially address but still unresolved. At line 97: "To better understand how individuals combine and integrate interactions ... " to me, the words "combine and integrate" recall this idea: Given we know how many neighbors the focal individual should consider, and what information is exchanged, what to do this information: summing? calculating averaging, taking the max, min? etc ... I do agree with what is written at line 174-178 (picking how many neighbors and how to pick are important). However, the words "combine and integrate" makes me think of what you do "after" the other two problems (picking the neighbors) are solved. In this specific situation, only one neighbor ends up being picked with a particular strategy, therefore there is not much left to "combine and integrate". I therefore only propose to make the statement more precise and aligned with what was actually done (it is possible that addressing the actual combination was either done in [14] or object of future work, I do not see how this can be simply assumed as averaging).

- Given the new details about the robotic experiments, I do forbid the use of the word "swarm" across the whole manuscript :-) For a group of robot to be considered a "swarm", embodied sensing is extremely important.

- I propose to replace Fig 14 with an actual finite state machine diagram, rather than a flow chart, because this would be more aligned with what explained in the main text (it would represent mainly the MOVE state and COMPUTE state, etc, the other being transitions across states).

Reviewer #3: I thank the authors for answering my previous concerns. I can recommend publication following some minor changes to the text (please see below).

Line 71 – Awkward sentence – please rephrase

Line 77 “In a previous work, we found experimental evidence that supports this assumption”

Line 81 ignore = “do not know”

Line 84 Awkward start to the sentence, please rephrase

Line 107 – 112: This reads more like the start of a discussion and should be removed from the end of the introduction.

**Have all data underlying the figures and results presented in the manuscript been provided?**

Reviewer #1: None

Reviewer #2: Yes

Reviewer #3: Yes

PLOS authors have the option to publish the peer review history of their article (what does this mean?). If published, this will include your full peer review and any attached files.

Reviewer #1: No

Reviewer #2: No

Reviewer #3: No

---

## [Editor Report · Decision Letter 2]

29 Jan 2020

Dear Dr. Theraulaz,

Thank you very much for submitting your manuscript "Computational and robotic modeling reveal parsimonious combinations of interactions between individuals in schooling fish" for consideration at PLOS Computational Biology. I considered the response to the previous review round satisfactory, and made a final review myself.  We are likely to accept this manuscript for publication, providing that you modify the manuscript according to the following recommendations.

Indeed, I believe that the paper still presents some minor issues that needs to be addressed before publication.

The major findings of the paper should be spelled out more clearly. At the current stage, the reader may get confused about what strategy is the best, and what value of k. To the best of my understanding, all strategies work well for k=2, leading to a quantitative match between experiments and simulations, but only the "most influential" strategy presents a behaviour qualitatively similar to the fish data for k=1. The robot experiments strengthen this aspect, providing a better (but not exact) quantitative match with the fish experiments for k=1. The way in which this message is passed in the current version is biased in favour of the k=1 match, which is mostly qualitative, providing in my opinion a spurious message. It actually seems that the best strategy to explain the data is "most influential" with k=1, but this is not true. Rather, k=1 most-influential neighbours is a sufficient number to get qualitative match, but k=2 neighbours are necessary to get a more quantitative match. I think that the abstract, summary, results and conclusions should be slightly rephrased.

I see that the data about k=3 for the MOST INFLUENTIAL strategy with robots are missing, and I couldn't find in the main text an explanation for this. This is a shortfall that must be clearly explained.

The rationale for choosing k=1 in the simulation about "collective motion of larger groups in an unbounded domain" should be clearly spelled out. I would have expected simulations also for k=2, which could show improved polarisation even for large groups. Although the behaviour resulting from k=1 is described sufficiently, the fact that with k=2 the same pattern arises is not guaranteed. Why no data has been included about k=2 for large groups?

In the conclusions, lines 651-663 are disconnected from the previous text, and should be better introduced. This paragraph has been included to motivate the biological plausibility of the proposed strategies. The important information is that identification of the most relevant neighbours does not require an explicit consideration of all the neighbours, which would be cognitively too demanding. It is suggested that this process is carried out in parallel by the forebrain, hence providing a cognitively plausible explanation of a strategy that would otherwise be very demanding. This should be clarified, possibly also discussing that the most salient stimuli could well be filtered in a way to automatically determine those that are most salient for the movement decision, i.e., suggesting a mechanisms for the computation of mostly influential individuals. This would somehow address the self-referentiality problem previously mentioned by the reviewers.

Minor issues:

lines 106-107: you may also mention collision avoidance

line 146: radius -> radii

lines 177-178: make a new sentence out of the brackets, or a footnote

[1] A letter containing a detailed list of your responses to all comments, and a description of the changes you have made in the manuscript. Please note while forming your response, if your article is accepted, you may have the opportunity to make the peer review history publicly available. The record will include editor decision letters (with reviews) and your responses to reviewer comments. If eligible, we will contact you to opt in or out

Sincerely,

Vito Trianni, Ph.D.

Guest Editor

PLOS Computational Biology

Samuel Gershman

Deputy Editor

PLOS Computational Biology

[LINK]

The paper still presents some minor issues that needs to be addressed before publication.

The major findings of the paper should be spelled out more clearly. At the current stage, the reader may get confused about what strategy is the best, and what value of k. To the best of my understanding, all strategies work well for k=2, leading to a quantitative match between experiments and simulations, but only the "most influential" strategy presents a behaviour qualitatively similar to the fish data for k=1. The robot experiments strengthen this aspect, providing a better (but not exact) quantitative match with the fish experiments for k=1. The way in which this message is passed in the current version is biased in favour of the k=1 match, which is mostly qualitative, providing in my opinion a spurious message. It actually seems that the best strategy to explain the data is "most influential" with k=1, but this is not true. Rather, k=1 neighbours is a sufficient number to get qualitative match, but k=2 is necessary to get a more quantitative match. I think that the abstract, summary, results and conclusions section should be slightly rephrased.

I see that the data about k=3 for the MOST INFLUENTIAL strategy with robots are missing, and I couldn't find in the main text an explanation for this. This is a shortfall that must be clearly explained.

The rationale for choosing k=1 in the simulation about "collective motion of larger groups in an unbounded domain" should be clearly spelled out. I would have expected simulations also for k=2, which could show improved polarisation even for large groups. Although the behaviour resulting from k=1 is described sufficiently, the fact that with k=2 the same pattern arises is not guaranteed. Why no data has been included about k=2 for large groups?

In the conclusions, lines 651-663 are disconnected from the rest, and should be better introduced. This paragraph has been included to motivate the biological plausibility of the proposed strategies. The important information is that identification of the most relevant neighbours does not require an explicit consideration of all the neighbours, which would be cognitively too demanding. It is suggested that this process is carried out in parallel by the forebrain. This should be clarified, possibly also discussing that the most salient stimuli could well be filtered in a way to automatically determine those that are most salient (i.e., mostly influential) for the movement decision. This would somehow address the self-referentiality problem previously mentioned by the reviewers.

Minor issues:

lines 106-107: you may also mention collision avoidance

line 146: radius -> radii

lines 177-178: make a new sentence out of the brackets, or a footnote
---

## [Editor Report · Decision Letter 3]

3 Feb 2020

Dear Dr. Theraulaz,

We are pleased to inform you that your manuscript 'Computational and robotic modeling reveal parsimonious combinations of interactions between individuals in schooling fish' has been provisionally accepted for publication in PLOS Computational Biology.

Before your manuscript can be formally accepted you will need to complete some formatting changes, which you will receive in a follow up email. A member of our team will be in touch within two working days with a set of requests.

Best regards,

Vito Trianni, Ph.D.

Guest Editor

PLOS Computational Biology

Samuel Gershman

Deputy Editor

PLOS Computational Biology

Thanks for the revised version. The new abstract represents properly the content of the paper. The extended discussion at the end of the paper has also improved a lot.

Note that, in my previous requests, I only raised the issue of a possible mis-interpretation from the point of view of a reader that does not know the details of the study. In fact, there was no misinterpretation of results from the authors' part. I had the impression that the wording used could lead to an higher emphasis on the k=1 aspects, rather than the k=2. Now, given that there is no error in the text and that you have improved the description of the robot results in the abstract, I would consider the issue closed.

---

## [Editor Report · Acceptance letter]

26 Feb 2020

PCOMPBIOL-D-19-01001R3

Computational and robotic modeling reveal parsimonious combinations of interactions between individuals in schooling fish

Dear Dr Theraulaz,

I am pleased to inform you that your manuscript has been formally accepted for publication in PLOS Computational Biology. Your manuscript is now with our production department and you will be notified of the publication date in due course.

With kind regards,

Laura Mallard
